# Retrieval is Not Enough: Enhancing RAG Reasoning through Test-Time Critique and Optimization

**Jiaqi Wei**[1,2]*  **Hao Zhou**[3]*  **Xiang Zhang**[4]*  **Di Zhang**[5]  **Zijie Qiu**[5]  **Wei Wei**[6]
**Jinzhe Li**[2,5]  **Wanli Ouyang**[2]  **Siqi Sun**[2,5]†

[1] Zhejiang University    [2] Shanghai Artificial Intelligence Laboratory
[3] South China University of Technology    [4] University of British Columbia
[5] Fudan University    [6] University of Hong Kong
jiaqi.wei@zju.edu.cn, siqisun@fudan.edu.cn

## Abstract

Retrieval-augmented generation (RAG) has become a widely adopted paradigm for enabling knowledge-grounded large language models (LLMs). However, standard RAG pipelines often fail to ensure that model reasoning remains consistent with the evidence retrieved, leading to factual inconsistencies or unsupported conclusions. In this work, we reinterpret RAG as *Retrieval-Augmented Reasoning* and identify a central but underexplored problem: *Reasoning Misalignment*—the divergence between an LLM's internal reasoning trajectory and the evidential constraints provided by retrieval. To address this issue, we propose ALIGNRAG, a novel iterative framework grounded in *Critique-Driven Alignment (CDA)*. We further introduce ALIGNRAG-AUTO, an autonomous variant that dynamically terminates refinement, removing the need to pre-specify the number of critique iterations. At the heart of ALIGNRAG lies a *contrastive critique synthesis* mechanism that generates retrieval-sensitive critiques while mitigating self-bias. This mechanism trains a dedicated retrieval-augmented *Critic Language Model (CLM)* using labeled critiques that distinguish between evidence-aligned and misaligned reasoning. Empirical evaluations show that our approach significantly improves reasoning fidelity. Our 8B-parameter CLM improves performance over the Self-Refine baseline by **12.1%** on out-of-domain tasks and outperforms a standard 72B-parameter CLM by **2.2%**. Furthermore, ALIGNRAG-AUTO achieves this state-of-the-art performance while dynamically determining the optimal number of refinement steps, enhancing efficiency and usability. ALIGNRAG remains compatible with existing RAG architectures as a *plug-and-play* module and demonstrates strong robustness under both informative and noisy retrieval scenarios. Overall, ALIGNRAG offers a principled solution for aligning model reasoning with retrieved evidence, substantially improving the factual reliability and robustness of RAG systems. **Our source code is provided at Github.**

## 1 Introduction

Large Language Models (LLMs) have significantly advanced natural language understanding and generation capabilities. Retrieval-Augmented Generation (RAG) [1–10] has emerged as a prominent paradigm for grounding LLM responses with external knowledge. However, RAG systems exhibit notable fragility, particularly when confronted with irrelevant or noisy retrieved evidence [11, 12]. Existing methods primarily rely on static, training-time optimizations, which are often insufficient to

---

*Equal contributions.
†Corresponding authors.

39th Conference on Neural Information Processing Systems (NeurIPS 2025).

address the dynamic challenges of *error propagation* during inference [13, 14]. We identify a critical, yet understudied, failure mode in RAG: *reasoning misalignment*—a disconnect between the model's reasoning process and the retrieved evidence. Prior work has focused predominantly on improving retrieval quality or generating more robust outputs, largely overlooking the explicit alignment of the reasoning steps with the provided evidence [15, 7, 16–18]. While reflective approaches like Self-RAG [19] attempt error detection, they often necessitate architectural modifications or task-specific fine-tuning, limiting their generalizability.

In this paper, we propose reconceptualizing RAG not merely as retrieval-augmented generation, but as *Retrieval-Augmented Reasoning*. We posit that RAG entails a structured reasoning process, typically involving stages such as (1) relevance assessment of retrieved documents, (2) mapping the query to specific points within the evidence, and (3) synthesizing evidence-integrated justifications. *Reasoning misalignment* occurs when breakdowns happen across these phases, for instance, when relevant evidence is retrieved but its content is not accurately integrated into the generated reasoning chain. These failure modes are pervasive and persist even with high-quality retrieval, remaining largely unaddressed by current methodologies.

To address reasoning misalignment, we introduce ALIGNRAG, a novel framework that employs *Critique-Driven Alignment (CDA)* to dynamically correct misalignments during inference using retrieval-augmented critiques. Distinct from general-purpose generation refinement techniques [20–23], ALIGNRAG incorporates a *contrastive critique synthesis* mechanism. This mechanism is designed to elicit evidence-grounded critiques and explicitly mitigate the self-bias commonly observed in self-critical LLMs [24–26, 21, 20]. This is achieved by training a dedicated retrieval-augmented *Critic Language Model (CLM)* on contrastive critiques. These critiques are generated by instruction-tuned LLMs and are guided by alignment signals derived from self-supervision or stronger external supervision. This paradigm breaks the circularity inherent in self-critical pipelines and specifically optimizes the CLM for evidence sensitivity, enabling it to reliably distinguish aligned from misaligned reasoning without propagating errors from potentially imperfect LLM feedback. At test time, ALIGNRAG iteratively refines the generated reasoning process by treating it as an optimizable artifact, transforming the RAG pipeline into an active reasoning system where critiques dynamically guide alignment with the retrieved evidence.

To enhance the framework's practicality, we also introduce ALIGNRAG-AUTO, a more autonomous variant that eliminates the need for manual tuning of iteration counts. By training the CLM to predict a special '[Good]' token upon generating a satisfactory response, ALIGNRAG-AUTO implements a dynamic stopping mechanism at inference time. This allows the system to terminate the refinement loop as soon as the reasoning is aligned with the evidence, saving substantial computational resources while maintaining high accuracy. This "hands-free" approach makes our framework more robust, efficient, and readily deployable across diverse tasks without domain-specific adjustments.

Extensive evaluations across seven benchmark datasets and three diverse model families firmly establish ALIGNRAG's state-of-the-art (SOTA) performance, consistently surpassing existing methods on a wide range of tasks. A key demonstration of its efficacy, driven by our critique learning strategy, is our 8B-parameter ALIGNRAG model outperforming a self-refine approach by 12.1% on out-of-domain (OOD) benchmarks and even a much larger vanilla 72B-parameter CLM by 2.2%. ALIGNRAG's robustness shines under both informative and noisy retrieval scenarios, proving that *when RAG retrieval falters,* ALIGNRAG *thrives*. Moreover, its design as a plug-and-play module ensures seamless integration into existing RAG pipelines without architectural modifications; for example, it enhanced InstructRAG's OOD accuracy by 9.4% when applied to the Qwen2.5-14B model. These comprehensive evaluations underscore ALIGNRAG's superiority and versatility for retrieval-augmented tasks, excelling in diverse retrieval conditions, ensuring high reasoning fidelity, and demonstrating strong generalization.

**In summary, this paper makes the following key contributions:** (1) We reconceptualize RAG as *Retrieval-Augmented Reasoning* and identify *Reasoning Misalignment* as a fundamental, understudied failure mode. (2) We introduce *critique learning* for RAG, a novel pipeline for training CLMs to generate retrieval-augmented critiques while mitigating self-preference bias through a contrastive synthesis approach. (3) We propose ALIGNRAG, a test-time framework that utilizes CDA steps to iteratively optimize the RAG reasoning process. We also present ALIGNRAG-AUTO, an autonomous extension that dynamically determines the optimal number of refinement steps, enhancing efficiency and usability. (4) We provide extensive empirical validation demonstrating that ALIGNRAG achieves

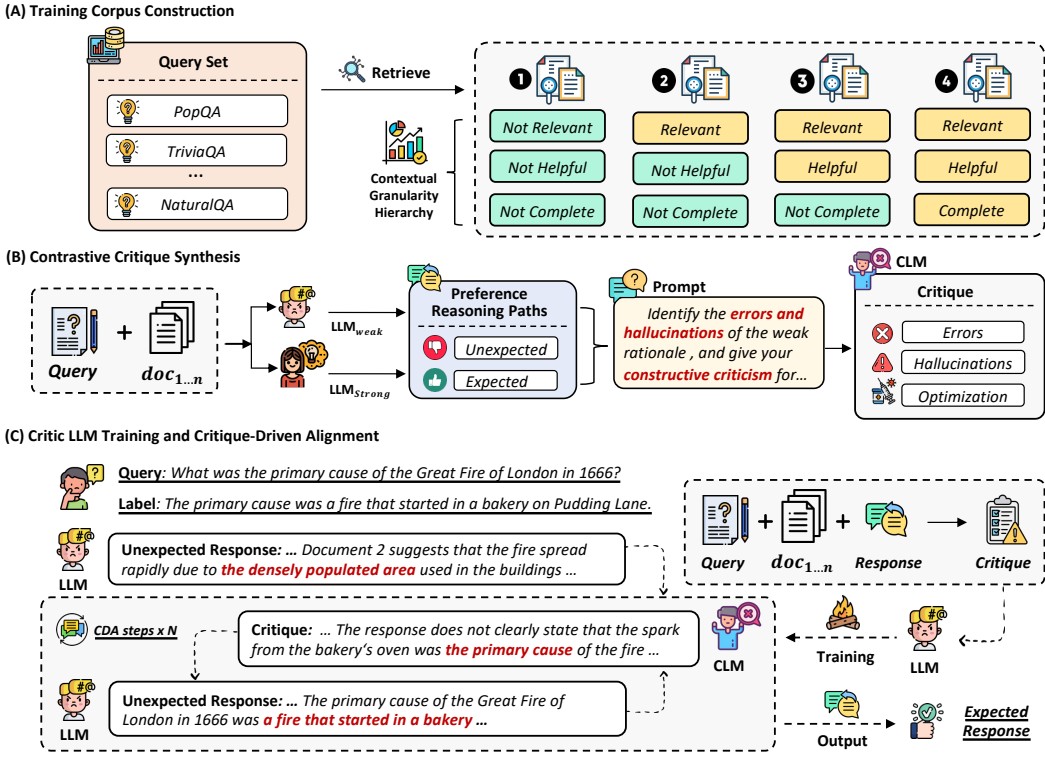

Figure 1: Overview of our ALIGNRAG Framework.

SOTA performance and significantly improves reasoning quality and robustness across various benchmarks and retrieval scenarios.

## 2 Reasoning Misalignment in RAG

RAG leverages external corpora $\mathcal{D}$ for grounded generation, with prior work largely focusing on improving retrieval [19] or generator robustness [13]. However, a critical and underexplored challenge is the *alignment* of the model's intrinsic reasoning process y with the specific evidential constraints from retrieved documents $\mathcal{D}$. Unlike error propagation in extended mathematical or code reasoning [14, 27], RAG failures often stem from inductive biases conflicting with external evidence, a qualitatively distinct problem.

We introduce *reasoning misalignment* as a novel, RAG-specific failure mode. It occurs when the model's constructed reasoning path y deviates significantly from the information or relationships within retrieved evidence $\mathcal{D}$, even when documents are relevant and contain the necessary facts. This is distinct from factual errors, context-free logical fallacies [28], or pure retrieval failures ($P(\mathcal{D} \mid q)$ issues). Instead, it signifies a breakdown in *faithful evidential integration*. Formally, misalignment is a structural deficiency in the conditional distribution $P(y \mid q, \mathcal{D})$, where y is the generated reasoning for query $q$ using $\mathcal{D}$. It is characterized by: (1) Erosion of Evidential Priors: Reasoning inconsistent with statistical or semantic properties in $\mathcal{D}$, such as $P(\text{span} \mid q, \mathcal{D})$ generated by the model differing significantly from the true salience. (2) Violation of Evidential Consistency: Deductive steps within y contradict logical inferences derivable from $\mathcal{D}$, i.e., $\mathcal{D} \not\models \text{step}_i$ for some reasoning step $\text{step}_i \in y$.

We decompose RAG reasoning into three interdependent phases, each susceptible to misalignment despite ideal retrieval:

*Phase 1: Relevance Assessment.* Misalignment occurs when the model fails to accurately gauge the relevance of specific text spans $s \subset d, d \in \mathcal{D}$ relative to the query $q$, effectively misestimating $P(s \text{ is relevant} \mid q, \mathcal{D})$ or assigning disproportionate weight to less relevant information [29].

*Phase 2: Query-Evidence Mapping.* This phase is susceptible when the model struggles to correctly identify how elements of $q$ map onto information in $\mathcal{D}$, representing a failure to correctly derive evidential relationships $\mathcal{R}(q, \mathcal{D})$ from $\mathcal{D}$ based on $q$ [30].

*Phase 3: Evidence-Integrated Synthesis.* Misalignment here involves generating reasoning steps or conclusions $\text{step}_i \in \text{y}$ that are not logically supported by the relevant evidence $E_i \subseteq \mathcal{D}$ used for that step, violating the entailment $E_i \models \text{step}_i$, or creating inconsistencies when synthesizing information from multiple parts of $\mathcal{D}$ into a coherent justification [31, 32, 14].

This taxonomy highlights reasoning misalignment as a pervasive issue orthogonal to retrieval quality or basic factuality. Static prompting cannot dynamically correct these evidence-grounded failures. To address this, we propose ALIGNRAG, a novel test-time framework enforcing evidential alignment via critique-guided alignment, offering a principled solution to this fundamental RAG challenge.

## 3 Critique-Driven Alignment for Retrieval-Augmented Reasoning

We present **Critique-Driven Alignment (CDA)**, a novel test-time refinement framework designed to mitigate reasoning misalignment in RAG. While conventional RAG pipelines often produce responses that partially or incorrectly reflect retrieved evidence, CDA introduces an explicit mechanism for identifying, diagnosing, and revising such failures via a learned critic model. This section details our approach, outlining the problem formulation (§3.1), our structured training methodology for critique learning (§3.2), and the iterative test-time alignment process (§3.3).

### 3.1 Problem Setting

Given an input query $q$ and a set of retrieved documents $\mathcal{D} = \{d_1, \ldots, d_n\}$, our objective is to refine an initial response $\text{y}_0 = \mathcal{M}_{\text{gen}}(q, \mathcal{D})$ through iterative critique-informed updates from a trained critic model $\mathcal{M}_{\text{critic}}$. To support the training of this critic model, we construct a critique supervision dataset:

$$\mathcal{S} = \{(q_i, a_i, \mathcal{D}_i, \mathbf{c}_i, \text{y}_{\text{exp},i}, \text{y}_{\text{unexp},i}, \Delta \text{y}_{\text{unexp},i})\}_{i=1}^N, \tag{1}$$

where each instance $i$ contains the query $q_i$, the ground-truth answer $a_i$, the retrieved documents $\mathcal{D}_i$, and a vector $\mathbf{c}_i = (\text{r}_i, \text{h}_i, \text{m}_i) \in \{0,1\}^3$ capturing the quality of the retrieved context along orthogonal axes: *relevance*, *helpfulness*, and *completeness* (related to explicit links between query elements and evidence). For each instance, we include a pair of responses $(\text{y}_{\text{exp},i}, \text{y}_{\text{unexp},i})$ generated by an expert strong and a weak model, respectively. These response pairs, combined with $q_i$ and $\mathcal{D}_i$, form a preference-augmented input representation $\mathcal{X}_{\text{pref},i} = (q_i, \mathcal{D}_i, \text{y}_{\text{exp},i}, \text{y}_{\text{unexp},i})$. Using $\mathcal{X}_{\text{pref},i}$ and potentially signals derived from $a_i$ and $\mathbf{c}_i$, we synthesize the supervision signal $\Delta \text{y}_{\text{unexp},i}$ representing a critique highlighting misalignments in $\text{y}_{\text{unexp},i}$ relative to $\mathcal{D}_i$, which is then used to train $\mathcal{M}_{\text{critic}}$.

### 3.2 Critic Training

#### 3.2.1 Training Corpus Construction

To model the ambiguity and diversity inherent in real-world retrieval scenarios [33–35], we construct a structured training dataset $\mathcal{S} = \{(q_i, a_i, \mathcal{D}_i, \mathbf{c}_i)\}_{i=1}^N$, where each instance includes a query $q_i$, its gold answer $a_i$, a retrieved document set $\mathcal{D}_i$, and a context granularity vector $\mathbf{c}_i \in \{0,1\}^3$. The vector encodes three orthogonal axes of contextual variation:

$$\mathbf{c}_i = (\text{r}_i, \text{h}_i, \text{m}_i), \tag{2}$$

where *Relevance* $\text{r}_i$ indicates the presence of relevant documents (derived from top-$k$ retrieval results, augmented with irrelevant documents sampled from unrelated queries). *Helpfulness* $\text{h}_i$ is a binary label reflecting whether the document(s) contain answer spans corresponding to $a_i$. *Completeness* $\text{m}_i$ is a document-set-level binary label signifying whether $\mathcal{D}_i$ collectively supports the full reasoning path required to derive $a_i$.

To systematically simulate varied degrees of answerability, we define a multiple-tier contextual granularity hierarchy (Fig. 1.A), exposing critic models to diverse evidence configurations and enabling fine-grained supervision. The details of data construction could refer to Appendix A.3.

#### 3.2.2 Contrastive Critique Synthesis

We propose **Contrastive Critique Synthesis (CCS)**, a **novel** methodology generating evidence-grounded feedback by contrasting reasoning trajectories from language models with differing capabilities. Since LLMs produce sequences from conditional distributions, $P(y|\text{input})$, self-critique risks

amplifying biases inherent in their own $P(y|\cdot)$. CCS counters this via a dedicated *Critic Language Model (CLM)* trained on structured contrastive pairs $\mathcal{X}_{\text{pref}} = (q, \mathcal{D}, y_{\text{exp}}, y_{\text{unexp}})$. This explicitly trains the CLM to identify misalignments w.r.t. $\mathcal{D}$, learning $P_{\text{critic}}(\text{critique}|\mathcal{X}_{\text{pref}})$ to capture deviations from evidential fidelity.

The CLM training employs a pairwise generation protocol. It samples an unexpert response $y_{\text{unexp}} \sim P_{\text{weak}}(y|q, \mathcal{D})$ (prone to misalignment) and an expert $y_{\text{exp}} \sim P_{\text{strong}}(y|q, \mathcal{D})$ (more aligned). To provide a structured input for learning the conditional distribution of critiques, we define a preference-augmented input tuple:

$$\mathcal{X}_{\text{pref}} = (q, \mathcal{D}, y_{\text{exp}}, y_{\text{unexp}}), \tag{3}$$

which conditions the CLM's learning on both desired trajectories ($y_{\text{exp}}$) and common failure patterns ($y_{\text{unexp}}$). This contrastive formulation offers two key benefits for training an effective critic: (1) it constrains the learning objective for $P_{\text{critic}}(\Delta y|\mathcal{X}_{\text{pref}})$ to focus on highlighting differences between $y_{\text{exp}}$ and $y_{\text{unexp}}$ that are grounded in $\mathcal{D}$, promoting evidence sensitivity and faithfulness [36]; and (2) it enables fine-grained diagnosis of specific misalignment types by analyzing path divergences [37].

The CLM learns $P_{\text{critic}}(\Delta y|\mathcal{X}_{\text{pref}})$ to generate a structured critique $\Delta y_{\text{unexp}}$, strictly grounded in $\mathcal{D}$. Training targets $\Delta y_{\text{unexp}}$ are synthesized via a Critique Function $\mathcal{F}$:

$$\Delta y_{\text{unexp}} = \mathcal{F}(\mathcal{X}_{\text{pref}}). \tag{4}$$

$\mathcal{M}_{\text{critic}}$ is trained to approximate $\mathcal{F}$. The final critique format $\Delta y_{\text{unexp}}$ is produced by an augmentation operator $\mathcal{G}[\text{output}(\mathcal{M}_{\text{critic}}(\mathcal{X}_{\text{pref}})), y_{\text{exp}}]$, which reformulates raw model output into constructive suggestions using $y_{\text{exp}}$ as reference. This framework enables the critic to localize and explain misalignments, providing high-quality feedback for RAG alignment.

### 3.2.3 Critic LLM Training

To instantiate the CLM, we introduce *Critique Fine-Tuning (CFT)* [38], a novel paradigm for training language models to produce constructive, evidence-grounded critiques. The goal is to transform a base model $\mathcal{M}_{\text{base}}$ into a proficient critic model $\mathcal{M}_{\text{critic}}$ using a synthetic dataset of critiques $\mathcal{C}$. Each training instance $\mathcal{C}_i \in \mathcal{C}$ is a tuple $(q, \mathcal{D}, y_{\text{unexp}}, \Delta y_{\text{unexp}}, y_{\text{exp}})$.

CFT formulates critique generation as a conditional sequence generation task. The training objective maximizes the likelihood of the model producing the correct critique $\Delta y_{\text{unexp}}$, conditioned on the full critique context $\mathcal{I}_{\text{critic}} = (q, \mathcal{D}, y_{\text{unexp}}, y_{\text{exp}})$. Formally, the objective is:

$$\mathcal{L}_{\text{CFT}}(\theta) = - \sum_{\mathcal{C}_i \in \mathcal{C}} \log p_\theta(\Delta y_{\text{unexp}} \mid \mathcal{I}_{\text{critic}}), \tag{5}$$

where $p_\theta$ is the probability distribution induced by the critic model $\mathcal{M}_{\text{critic}}$, parameterized by $\theta$. This formulation enables the model to generate actionable, targeted feedback that improves downstream response quality. By decoupling critique generation from the target model's own outputs, CFT mitigates the self-preference bias commonly observed in iterative self-correction methods.

### 3.3 Critique-Driven Alignment Systems

To address reasoning misalignment in RAG at inference, we propose Critique-Driven Alignment (CDA). Unlike standard single-pass RAG generation, $y_0 = \mathcal{M}_{\text{gen}}(q, \mathcal{D})$:

$$y_0 = \mathcal{M}_{\text{gen}}(q, \mathcal{D}), \tag{6}$$

CDA reconceptualizes inference as an iterative optimization over a latent reasoning space $\mathcal{Y}$. Each iteration leverages critiques to incrementally improve output alignment with retrieved evidence.

A learned critic $\mathcal{M}_{\text{critic}}$ iteratively analyzes intermediate generations and provides critiques suggesting improvements. This yields a refinement trajectory:

$$y_0 \xrightarrow{\text{CDA}} y_1 \xrightarrow{\text{CDA}} \cdots \xrightarrow{\text{CDA}} y_T, \tag{7}$$

where each transition $y_t \to y_{t+1}$ is critique-guided. At step $t < T$, the critic outputs an edit signal $\Delta y_t$ identifying issues in $y_t$ and proposing $\mathcal{D}$-grounded revisions. $y_{t+1}$ is generated by augmenting $\mathcal{M}_{\text{gen}}$'s input:

$$y_{t+1} = \mathcal{M}_{\text{gen}}(y_t \oplus \Delta y_t), \tag{8}$$

where $\oplus$ denotes prompt augmentation with critique feedback. $\Delta \mathrm{y}_t$ acts as a pseudo-gradient in discrete space, directing the generator toward $\mathcal{D}$-aligned reasoning.

The final CDA output is the trajectory's terminal state:

$$\mathrm{y}_{\mathrm{final}} = \mathrm{CDA}(q, \mathcal{D}) := \mathrm{y}_T. \tag{9}$$

This framework elevates alignment from static supervision to a dynamic iterative process at test-time, enabling demonstrably more reliable and evidence-grounded reasoning than existing RAG.

### 3.4 AlignRAG-auto: Dynamic and Domain-General Alignment

We also develop **AlignRAG-auto**, an autonomous extension of CDA that eliminates the need for manual iteration tuning and demonstrates strong cross-domain generalization. Whereas the baseline CDA requires specifying the maximum number of refinement steps $T$, AlignRAG-auto leverages a lightweight control mechanism that dynamically determines when alignment has been achieved. This improves both efficiency and robustness at deployment.

#### 3.4.1 Training for Dynamic Refinement

We modify the training process for the Critic Language Model (CLM) to support dynamic inference. For each "unexpected" response $y_{\mathrm{unexp}}$, we first determine correctness relative to ground truth. Incorrect responses are labeled with a `[Bad]` token, while correct ones are labeled with `[Good]`. The CLM is then trained to generate the appropriate control token followed by a structured critique:

$$p_\theta(\,[\texttt{Good/Bad}]\,, \Delta y \mid q, \mathcal{D}, y_{\mathrm{unexp}}). \tag{10}$$

This dual-target training objective equips the critic with both evaluative and corrective capacity. By conditioning critiques on explicit correctness judgments, the CLM learns to terminate refinement as soon as sufficient alignment is achieved.

#### 3.4.2 Dynamic Inference-Time Stopping

At inference, AlignRAG-auto proceeds iteratively as in CDA but introduces a dynamic stopping rule. If the critic predicts `[Good]`, the system halts refinement and accepts the current candidate:

$$y_{t+1} = \begin{cases} y_t, & \text{if CLM outputs } [\texttt{Good}] \\ \mathcal{M}_{\mathrm{gen}}(y_t \oplus \Delta y_t), & \text{if CLM outputs } [\texttt{Bad}]. \end{cases} \tag{11}$$

This adaptive strategy avoids unnecessary iterations, saving compute while maintaining high fidelity to retrieved evidence. Empirically, we observe that only a subset of responses require multiple refinements, yielding substantial runtime reductions without compromising accuracy (see Table 2).

## 4 Experiments

### 4.1 Experiment Setup

We evaluate our method using three instruction-tuned backbones: Qwen2.5-7B-Instruct [39], Qwen2.5-14B-Instruct [39], and LLaMA3.1-8B-Instruct [40]. For simplicity, we refer to them as Qwen2.5-7B, Qwen2.5-14B, and LLaMA3.1-8B.

**Dataset.** To train a strong critique generator, we construct a 10K dataset by sampling 2K instances from each of five benchmarks: PopQA [41], TriviaQA [42], NaturalQuestions [43], 2WikiMulti-hopQA [44], and ASQA [45]. Furthermore, we evaluate our method on the same five in-domain benchmarks, along with two out-of-distribution (OOD) tasks, *i.e.*, HotpotQA [46] and SQuAD [47].

**Baselines.** In our experiments, we compare our method against a range of non-retrieval and retrieval-based baselines. For non-retrieval baselines, we include Chain-of-Thought (CoT) prompting [48, 49] applied to models without retrieval augmentation. For standard RAG, we report performance from Vanilla Reasoning [13, 50], which performs step-by-step answer generation based on the retrieved passages. To assess the benefits of intermediate supervision, we include training-time refinement baselines such as RetRobust[33] and InstructRAG[13]. **Our main comparison is with these test-time refinement methods, as they share similar objectives.** For test-time refinement, we evaluate Self-RAG [19] and Self-Refine, which iteratively revises outputs based on self-generated critique.

Table 1: Overall performance comparison of RAG systems employing various knowledge refinement strategies and reasoning configurations across five question-answering (QA) benchmarks. To ensure a fair evaluation, all systems are tested under a **single-iteration test-time refinement** setting. Results marked with * are reproduced from [13]. Missing results in the original paper are denoted by "–". To highlight our method's **impact of different model backbones**, we use the following color-coded notation for performance improvements: ($\triangle$) represents the Qwen-2.5-Instruct$_{7B}$, ($\triangle$) represents the Qwen-2.5-Instruct$_{14B}$, and ($\triangle$) represents the Llama-3-Instruct$_{8B}$.

| Method | NQ | MultiHopQA | TriviaQA | PopQA | ASQA | Avg. |
| Metric | accuracy | accuracy | accuracy | accuracy | str-em | |
|---|---|---|---|---|---|---|
| *Baselines w/o Retrieval* | | | | | | |
| **Chain-of-thought** [48] | | | | | | |
| Qwen-2.5-Instruct$_{7B}$ | 33.9 | 45.0 | 58.3 | 26.9 | 20.5 | 36.9 |
| Qwen-2.5-Instruct$_{14B}$ | 48.1 | 49.3 | 72.8 | 25.4 | 31.6 | 45.4 |
| Llama-3.1-Instruct$_{8B}$ | 42.1 | 41.9 | 61.8 | 26.9 | 25.1 | 40.0 |
| *Standard RAG with Reasoning* | | | | | | |
| **Vanilla Reasoning** | | | | | | |
| Qwen-2.5-Instruct$_{7B}$ | 60.2 | 44.7 | 73.2 | 63.7 | 42.8 | 56.9 |
| Qwen-2.5-Instruct$_{14B}$ | 63.6 | 44.8 | 77.0 | 65.3 | 45.2 | 59.2 |
| Llama-3.1-Instruct$_{8B}$ | 62.0 | 43.0 | 73.4 | 65.0 | 45.2 | 57.7 |
| *RAG w/ Training-time Refinement* | | | | | | |
| **RetRobust** [33] | | | | | | |
| Llama-2$_{13B}$* | 39.6 | 51.5 | – | – | – | – |
| Llama-3-Instruct$_{8B}$* | 54.2 | 54.7 | 71.5 | 56.5 | 40.5 | 55.5 |
| **InstructRAG** [13] | | | | | | |
| Qwen-2.5-Instruct$_{7B}$ | 63.8 | 46.3 | 76.1 | 67.5 | 47.5 | 60.2 |
| Qwen-2.5-Instruct$_{14B}$ | 66.3 | 47.3 | 78.7 | 67.8 | 48.5 | 61.7 |
| Llama-3.1-Instruct$_{8B}$ | 66.3 | 45.1 | 76.6 | 66.9 | 47.2 | 60.4 |
| **RAG w/ Test-time Refinement** | | | | | | |
| **Self-RAG** [19] | | | | | | |
| Llama-2$_{7B}$ + CLM$_{7B}$* | 42.4 | 35.9 | 68.9 | 55.8 | 30.0 | 46.6 |
| Llama-2$_{13B}$ + CLM$_{13B}$* | 46.4 | 36.0 | 70.4 | 56.3 | 31.4 | 48.1 |
| Llama-3-Instruct$_{8B}$ + CLM$_{8B}$* | 42.8 | 32.9 | 71.4 | 55.8 | 36.9 | 48.0 |
| **Self-Refine** | | | | | | |
| Qwen-2.5-Instruct$_{7B}$ + SELF$_{7B}$ | 61.6($\triangle$) | 45.0($\triangle$) | 74.4($\triangle$) | 65.5($\triangle$) | 45.2($\triangle$) | 58.3($\triangle$) |
| Qwen-2.5-Instruct$_{14B}$ + SELF$_{14B}$ | 65.1($\triangle$) | 46.1($\triangle$) | 78.0($\triangle$) | 67.0($\triangle$) | 47.3($\triangle$) | 60.7($\triangle$) |
| Llama-3.1-Instruct$_{8B}$ + SELF$_{8B}$ | 61.4($\triangle$) | 42.8($\triangle$) | 74.1($\triangle$) | 66.1($\triangle$) | 44.7($\triangle$) | 57.8($\triangle$) |
| **AlignRAG-fixed** | | | | | | |
| Qwen-2.5-Instruct$_{7B}$ + CLM$_{8B}$ | 65.9 (↑ 4.3%) | 49.5 (↑ 4.5%) | 77.8 (↑ 3.4%) | 68.4 (↑ 2.9%) | 48.9 (↑ 3.7%) | 62.1 (↑ 3.8%) |
| Qwen-2.5-Instruct$_{14B}$ + CLM$_{8B}$ | 67.7 (↑ 2.6%) | 49.8 (↑ 3.7%) | 79.5 (↑ 1.5%) | 68.4 (↑ 1.4%) | 48.6 (↑ 1.3%) | 62.8 (↑ 2.1%) |
| Llama-3.1-Instruct$_{8B}$ + CLM$_{8B}$ | 65.3 (↑ 3.9%) | 47.0 (↑ 4.2%) | 77.0 (↑ 2.9%) | 66.5 (↑ 0.4%) | 47.1 (↑ 2.4%) | 60.6 (↑ 2.8%) |

**Evaluation metrics.** Following previous work [51], we adopt the official correctness metric (*str-em*) for ASQA [45], and use *accuracy* for the other tasks, which measures whether the final generations of the model align with the ground-truth [41, 52].

**Implementation Details.** For the CLM, we adopt LLaMA3.1-8B-Instruct as the backbone and fine-tune it using LoRA for parameter-efficient training. Moreover, the *strong LLM* we use to generate *expected* responses is LLaMA3.1-8B-Instruct, and the *weak LLM* we use to generate *unexpected* responses is Qwen2.5-0.5B-Instruct [39]. We set the retrieval Top-$K$ to 5 for each question.

## 4.2 Main Result

Table 1 shows the overall performance of our method and the baselines in various families and sizes of the base model on five benchmarks. And we provide all the additional results in Appendix A.6

First, compared to non-retrieval baselines such as Chain-of-Thought (CoT) prompting, all retrieval-augmented methods achieve significantly better performance, demonstrating the importance of incorporating relevant external knowledge. Second, we observe further gains when applying training-time refinement methods. In particular, InstructRAG achieves strong performance across all backbones, outperforming Vanilla RAG by a large margin, confirming the value of training refinement strategies.

Notably, ALIGNRAG achieves the best overall results on all three backbones compared to other test-time refinement methods. It surpasses Self-RAG and Self-Refine by notable margins, achieving an average accuracy of 62.8% compared to 48.1% and 60.7%, respectively. The performance improvement is consistent across all benchmarks, highlighting both the effectiveness and the strong generalization of our approach. This demonstrates that our critique-driven alignment strategy can better guide the reasoning process and overcome the limitations of purely self-generated feedback.

**Analysis of AlignRAG-auto.** The experimental results, detailed in Table 2, reveal a compelling comparison between the fixed-iteration and autonomous alignment strategies. Notably, **AlignRAG-auto** consistently achieves performance on par with, and in many cases slightly superior to, its

**AlignRAG-fixed** counterpart across all datasets and model sizes. For instance, on the NQ and ASQA benchmarks, AlignRAG-auto demonstrates clear improvements, suggesting its dynamic termination mechanism is highly effective.

This is a crucial finding: the autonomous variant **does not sacrifice accuracy for efficiency**. Instead, it demonstrates that the Critic Language Model is well-calibrated to dynamically determine the optimal number of refinement steps, leading to robust performance without the need for manual hyperparameter tuning. This validates AlignRAG-auto as a more practical and efficient framework for real-world deployment.

Figure 2: Performance Comparison: AlignRAG-fixed (1 iter.) vs. AlignRAG-auto

| Dataset | AlignRAG-fixed (1 iter.) | | | AlignRAG-auto | | |
|---|---|---|---|---|---|---|
| | **8B** | **7B** | **14B** | **8B** | **7B** | **14B** |
| PopQA | 66.5 | 68.4 | 68.4 | 67.6 | 68.1 | 68.3 |
| TriviaQA | 77.0 | 77.8 | 79.5 | 77.6 | 78.1 | 79.9 |
| NQ | 65.3 | 65.9 | 67.7 | 66.8 | 67.3 | 69.0 |
| 2WikiMultiHopQA | 47.0 | 49.5 | 49.8 | 47.6 | 49.3 | 50.2 |
| ASQA | 47.1 | 48.9 | 48.6 | 48.8 | 49.6 | 49.8 |

### 4.3 Analysis

**Note: All subsequent experimental analyses are based on AlignRAG-fixed. For convenience, we'll use AlignRAG instead.**

**Generalization to OOD Scenarios.** To assess the generalization capability of our method beyond the domains seen during training, we conduct out-of-distribution (OOD) evaluations on two widely-used and challenging benchmarks that are held out from the training set, *i.e.*, HotpotQA [46] and SQuAD [47]. This evaluation setting enables us to examine how well the model transfers its reasoning and alignment abilities to novel domains. As shown in Figure 3, we compare ALIGNRAG with two baselines. ALIGNRAG consistently achieves the lowest performance drop across all backbones, outperforming both baselines by a large margin. For example, on LLaMA3.1-8B, ALIGNRAG reduces the performance drop to 32.4% compared

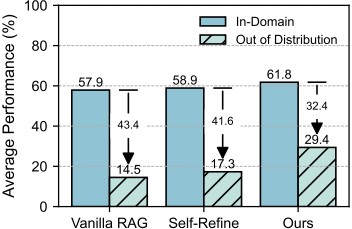

Figure 3: Drop in average OOD performance compared to average In-Domain performance.

to 41.6% for Self-Refine. These results demonstrate that our CDA mechanism not only improves in-domain reasoning but also enhances robustness under domain shift.

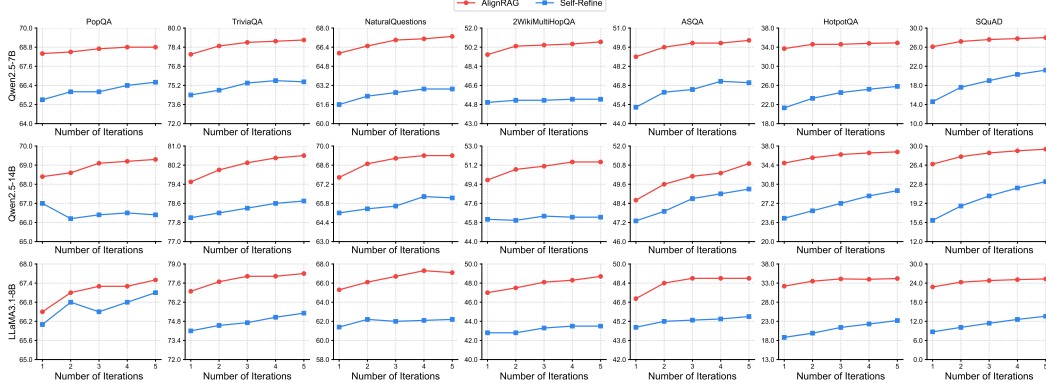

Figure 4: Performance comparison of ALIGNRAG and Self-Refine across five refinement iterations.

**Scalable Test-time Reasoning via Iterative Alignment Strategy.** To assess test-time scalability of different refinement methods, we plot accuracy over five refinement steps across seven benchmarks, comparing the performance of ALIGNRAG with the Self-Refine baseline (Figure 4). The curves reveal two notable trends. First, both methods generally benefit from iterative alignment, with accuracy improving on most tasks as the number of refinement steps increases. This indicates that reasoning can scale with additional refinement steps. However, we occasionally observe slight degradation beyond a certain point, which we attribute to potential noise accumulation or overcorrection during excessive iterations. Second, ALIGNRAG consistently outperforms Self-Refine across all iterations

and benchmarks with notable margins. These findings demonstrate that ALIGNRAG not only enables scalable reasoning but also provides more stable and robust improvements.

**When RAG Retrieval Falters, ALIGNRAG Thrives.**   RAG systems are susceptible to substantial performance degradation when retrieved documents lack pertinent answers, a prevalent yet insufficiently characterized failure mode termed *Noisy Retrieval*. Even in the absence of explicit answers within the retrieved corpus, ALIGN-RAG adeptly filters distractors to isolate underlying reasoning signals while concurrently activating complementary parametric knowledge inherent to the base model.

This dual strategy, as demonstrated in Figures 5a and 5b, empowers robust reasoning even under these adversarial conditions. Consequently, ALIGNRAG (Figure 5a) significantly outperforms conventional methods like Vanilla RAG and Self-Refine, which falter in such scenarios, by effectively transforming imperfect external retrievals and inherent model understanding into actionable insights.

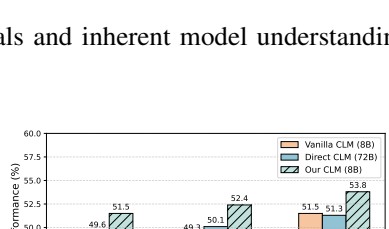

(a) w/o answer.          (b) w/ answer.

Figure 5: ALIGNRAG performance under *Noisy Retrieval* (a) and *Informative Retrieval* (b) scenarios.

**Superior Alignment Guidance Surpassing Strong LLM Baselines.**   We examine the effectiveness of using a high-capacity *Strong LLM*, *i.e.*, Qwen2.5-72B-Instruct, in our Contrastive Critique Synthesis (CCS) framework. Specifically, we compare three CLM supervision strategies under three backbones: (1) *Vanilla Critique Synthesis*, where the CLM is trained with critiques directly generated by Qwen2.5-72B-Instruct, (2) directly using the 72B model as the CLM, and (3) our CCS-based CLM trained with critiques synthesized by the 72B *Strong LLM*.

Figure 6: Comparison of different CLM supervision strategies.

As shown in Figure 6, our method consistently outperforms vanilla critique synthesis, validating the benefit of introducing contrastive reasoning signals. Remarkably, our method even surpasses the directly supervised 72B CLM in all backbones, suggesting that contrastive critique training enhances generalization and reduces reliance on model scale. These results demonstrate the strength of our approach in leveraging powerful LLMs for scalable and efficient CLM training.

**Integrate as a Plug-and-play Module into Existing RAG Pipelines.**   To evaluate the generality and plug-and-play nature of our method, we integrate it into the InstructRAG framework across three backbones. Table 2 reports the performance under both In-Domain (ID) and Out-of-Domain (OOD) evaluation. We observe consistent improvements in both familiar and unseen distributions. The variant with alignment significantly outperforms the original InstructRAG [13], demonstrating that our method can be incorporated into existing RAG pipelines in a zero-modification, test-time manner, highlighting its strong compatibility and practical utility.

Table 2: Combination of training-time (InstructRAG) and test-time alignment.

| Method | ID (avg.) | OOD (avg.) |
|---|---|---|
| *Qwen2.5-7B* | | |
| InstructRAG | 59.5 ($\Delta$) | 28.0 ($\Delta$) |
| w/ Alignment | 61.5 (↑ 2.0%) | 30.1 (↑ 2.1%) |
| w/ Alignment[1] | 63.0 (↑ 3.5%) | 31.7 (↑ 3.7%) |
| *Qwen2.5-14B* | | |
| InstructRAG | 61.7 ($\Delta$) | 24.9 ($\Delta$) |
| w/ Alignment | 62.5 (↑ 0.8%) | 33.4 (↑ 8.5%) |
| w/ Alignment[1] | 63.9 (↑ 2.2%) | 34.3 (↑ 9.4%) |
| *LLaMA3.1-8B* | | |
| InstructRAG | 60.4 ($\Delta$) | 28.4 ($\Delta$) |
| w/ Alignment | 61.7 (↑ 1.3%) | 29.4 (↑ 1.0%) |
| w/ Alignment[1] | 61.9 (↑ 1.5%) | 30.5 (↑ 2.1%) |

## 4.4   Ablation Study

---

[1]CLM trained with critiques synthesized by Qwen2.5-72B-Instruct.

We perform an ablation study on seven benchmarks to isolate the effect of the Critic Language Model (CLM) and our contrastive critique synthesis (CCS). Specifically, we compare four settings: (1) **Vanilla RAG**, which generates without critique; (2) **RAG + Frozen CLM**, where a pretrained LLaMA3.1-8B serves as an untrained critic; (3) **RAG + Trained CLM**, where the CLM is fine-tuned on critiques over weak LLM responses, but without contrastive signal; (4) **RAG + Trained CLM (CCS)**, our full method with contrastive critique synthesis.

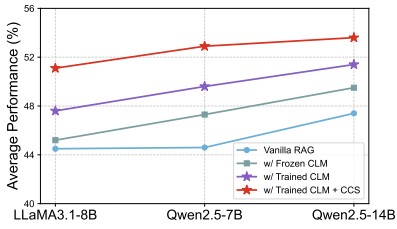

Figure 7: Ablation study on the CLM and our CCS.

Figure 7 shows that even a frozen CLM consistently improves over Vanilla RAG, confirming the utility of auxiliary critique. Training the CLM with weak LLM critiques yields further gains by capturing recurring error patterns. The largest improvements come from CCS, which explicitly contrasts expert and weak responses. For example, on Qwen2.5-7B, CCS raises average accuracy from 49.6% to 52.9%. Complete results are presented in Table 4 (Appendix).

## 5 Related Work

RAG [53–59] enhances LLMs by grounding generation in external knowledge. While significant efforts have focused on improving retrieval accuracy [60–64] or training better generators [65–68], a persistent challenge is the presence of noisy or irrelevant retrieved content. To mitigate this, recent methods filter or denoise context, predominantly via *training-time* optimizations [15, 69–72], such as InstructRAG's [13] self-supervised denoising. However, these approaches are limited by their static nature and fail to address dynamic error propagation during inference [14].

Even with accurate retrieval, the generated reasoning may deviate from the evidence. We identify this *reasoning misalignment* as a critical, underexplored RAG failure mode. Prior attempts like Self-RAG [19] introduce special tokens to control reasoning but require architectural modifications. In contrast, our novel Critique-Driven Alignment (CDA) is a test-time method that dynamically realigns reasoning with evidence *without modifying the base model architecture*, offering a plug-and-play alternative to training-heavy variants. Furthermore, unlike general self-refinement techniques [73–75], we propose a novel *critique learning* paradigm training a dedicated Critic Language Model (CLM) specifically for evidence-grounded critiques. Crucially, this paradigm explicitly mitigates the self-bias inherent in self-critiquing LLMs [24–26, 21, 20]. Unlike external verification methods [76, 77] that act as post-hoc filters, our trained CLM actively optimizes the evidence-grounded reasoning process as an artifact during inference, enabling dynamic alignment with retrieved knowledge.

## 6 Conclusion

We present ALIGNRAG, an iterative framework that reframes Retrieval-Augmented Generation as *Retrieval-Augmented Reasoning* to tackle the overlooked challenge of *Reasoning Misalignment*. Its key innovation, *Critique-Driven Alignment (CDA)*, uses a specialized *Critic Language Model (CLM)*—trained via *contrastive critique synthesis*—to boost evidence sensitivity and correct reasoning errors at inference. ALIGNRAG sets new state-of-the-art. This principled approach advances the reliability and faithfulness of retrieval-augmented systems.

## Acknowledgements

This project was fully supported by the Shanghai Artificial Intelligence Laboratory (S.S.). This work was done during Jiaqi Wei's internship at Shanghai Artificial Intelligence Laboratory.

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

# A Appendix

## A.1 Limitations

While ALIGNRAG represents a significant step forward in retrieval-augmented reasoning through critique-guided optimization, several limitations remain. Although the framework improves robustness to noisy or partially irrelevant retrieved content, its effectiveness may diminish under extreme retrieval failure, specifically, when none of the retrieved documents are relevant to the query. In such cases, the quality of retrieval-augmented critiques deteriorates, as they are inherently dependent on the informativeness and accuracy of the retrieved evidence. Consequently, the CLM's capacity to steer the generator toward faithful reasoning may be constrained. The interaction between retrieval quality and the model's corrective ability remains an open area for further exploration. Moreover, despite being designed as a plug-and-play module, optimal integration of ALIGNRAG may require minor tuning depending on the generator LLM and the underlying RAG architecture. Standardizing adaptation protocols for deploying the CLM across diverse pipelines could enhance its ease of adoption and generalizability. Despite these limitations, ALIGNRAG establishes a robust foundation for improving evidence alignment in RAG systems and opens promising avenues for future research.

## A.2 Additional Implementation Details

**Retrieve Setup.** We use the Wikipedia corpus provided by [78] as the default external knowledge source for retrieval. We evaluate our method on seven diverse QA benchmarks spanning multiple task types, including standard factoid QA, multi-hop reasoning, and long-form generation. PopQA [41], TriviaQA [42], NaturalQuestions [43], and SQuAD [47] fall under standard factoid QA, where models answer factual questions based on Wikipedia or web-based evidence.

- PopQA focuses on entity-centric questions derived from structured knowledge bases, testing factual recall over encyclopedic content.
- TriviaQA contains trivia-style questions authored by enthusiasts, each paired with multiple distant-supervised evidence documents.
- NaturalQuestions presents real user queries issued to Google Search, with answers extracted from Wikipedia, simulating realistic search behavior.
- ASQA [45] is a long-form QA benchmark focused on ambiguous questions with paragraph-level answers.
- 2WikiMultiHopQA [44] and HotpotQA [46] are multi-hop QA datasets that require reasoning over multiple passages. 2WikiMultiHopQA evaluates compositional reasoning across two Wikipedia articles, while HotpotQA incorporates both supporting and distracting sentences, encouraging interpretable multi-step reasoning.
- SQuAD is a widely used extractive QA dataset where answers are short spans from Wikipedia passages.

Following the setup in InstructRAG [13], we adopt dataset-specific retrievers for each query: Contriever-MS MARCO for PopQA and TriviaQA, DPR for NaturalQuestions, GTR for ASQA, and BM25 for 2WikiMultiHopQA. For HotpotQA and SQuAD, we adopt the `e5-base-v2` encoder. By default, we retrieve the top 5 most relevant documents from Wikipedia corpus for each question.

**Training Details.** We fine-tune our models using the LoRA method on 2 NVIDIA A100 GPUs, each with 80GB of memory. The fine-tuning process is conducted over 2 epochs with a learning rate of 1e-5 using the AdamW optimizer and employs a per-device batch size of 16, leveraging gradient accumulation to handle larger effective batch sizes. We set the LoRA-specific hyperparameters as follows: $lora\_rank = 16$ and $lora\_alpha = 64$, ensuring efficient adaptation to downstream tasks. The sequence cutoff length is 6144 tokens, with a warmup ratio of 0.1 applied to stabilize training. Additionally, we utilize bf16 (brain floating point) precision to reduce memory usage and accelerate training while maintaining numerical stability.

## A.3 Training Corpus Construction Details

To systematically simulate varying degrees of answerability, we introduce a novel four-tier *Contextual Granularity Hierarchy* (Figure 1), which forms the basis for structured context-aware critique learning.

This hierarchy is designed to expose critique models to a broad spectrum of evidence conditions, thereby facilitating fine-grained supervision under explicitly controlled scenarios.

The hierarchy is defined along three orthogonal dimensions of contextual variation—*relevance*, *helpfulness*, and *completeness*—and comprises the following four levels:

*Hierarchy-1: Not Relevant, Not Helpful, Not Complete.* We sample 200 instances per benchmark, where the context is randomly selected from evidence retrieved for unrelated questions. These contexts are neither topically relevant nor contain partial answers, thus offering no utility in addressing the question.

*Hierarchy-2: Relevant, Not Helpful, Not Complete.* We sample 400 instances per benchmark, in which the retrieved context comprises the top-5 documents relevant to the question but lacking any content that supports a correct answer. Although relevant, these contexts remain unhelpful and incomplete.

*Hierarchy-3 & 4: Relevant, Helpful, Not Complete / Complete.* We sample 1,400 instances per benchmark where the context is both relevant and helpful, containing either partial (no single document provides a complete answer) or complete answer-supporting information. To capture varying levels of difficulty, we categorize queries into five tiers based on the number of documents that individually contain supporting evidence (ranging from 1 to 5). Easier queries correspond to a higher number of such documents. We sample 400, 400, 200, 200, and 200 instances across these five levels, respectively.

This hierarchical corpus introduces a novel fine-grained supervision signal for training critique models, enabling a more nuanced understanding of answerability and evidence quality in retrieval-augmented generation.

## A.4 Critic LLM Training via CPO

In addition to *Critique Fine-Tuning (CFT)*, we introduce a novel training paradigm for critique language models, termed **Critique Preference Optimization (CPO)**. CPO extends the Direct Preference Optimization (DPO) framework [79] to the domain of critique generation, enabling preference-based alignment of critique models with respect to human-quality judgments.

For each training example, we construct a pair of candidate critiques: a *rejected* critique $\Delta y_{\text{unexp}}^-$ generated by a weaker model $\mathcal{M}_{\text{weak}}$, and an *accepted* critique $\Delta y_{\text{unexp}}^+$ from a stronger model $\mathcal{M}_{\text{strong}}$. The critic model $\mathcal{M}_{\text{critic}}$ is then optimized to prefer the stronger critique over the weaker one using a ranking-based objective:

$$\mathcal{L}_{\text{CPO}} = -\mathbb{E}_{\mathcal{C}} \left[ \log \sigma \left( \beta \log \frac{p_\theta(\Delta y_{\text{unexp}}^+ \mid q, \mathcal{D}, y_{\text{unexp}}^+)}{p_\theta(\Delta y_{\text{unexp}}^- \mid q, \mathcal{D}, y_{\text{unexp}}^+)} \right) \right], \tag{12}$$

where $\sigma(\cdot)$ denotes the sigmoid function, $\beta$ is a temperature parameter controlling preference sharpness, and $p_\theta$ is the conditional likelihood of a critique under the model. Importantly, the conditioning includes the stronger generation $y_{\text{unexp}}^+$ to ground the critique in high-quality reference behavior.

This training strategy represents a novel application of preference optimization to the critique generation setting. It allows the model to learn fine-grained distinctions in critique quality and improves alignment with human preferences, surpassing traditional supervised learning approaches in adaptability and scalability.

## A.5 Pseudo-code of Novel Algorithms for Critique-Aware Learning

To promote clarity and reproducibility, we present formalized pseudo-code for the core contributions of our framework, highlighting novel procedures for critique generation, fine-tuning, and alignment. These algorithmic components reflect our key innovations in critique-aware generation and optimization.

Algorithm 1 introduces **Contrastive Critique Synthesis**, a novel mechanism that elicits actionable critiques by contrasting outputs from a weak and a strong model. This facilitates the identification of failure modes in weaker generations using preference-informed critique models. Algorithm 2, **Critique Fine-Tuning (CFT)**, formalizes a supervised learning regime using synthetic critiques and structured input templates to fine-tune a base model toward producing useful critiques.

In Algorithm 3, we present **Critique Preference Optimization (CPO)**, which extends the Direct Preference Optimization (DPO) framework to critique generation. This formulation enables preference-based alignment of critique models using pairs of more and less preferred critiques. Lastly, Algorithm 4 describes **Critique-Driven Alignment (CDA)**, a novel iterative refinement procedure that integrates critique signals into the generation loop, producing responses that are successively improved based on model-generated feedback.

Collectively, these algorithmic components define a unified, modular framework for critique-aware alignment, marking a novel contribution to controllable and preference-aligned language model training.

**Algorithm 1** CONTRASTIVE CRITIQUE SYNTHESIS (Novel critique generation via response comparison)

---

**Require:** Input query $q$, contextual grounding $\mathcal{D}$, weak model $\mathcal{M}_{\text{weak}}$, strong model $\mathcal{M}_{\text{strong}}$, critique model $\mathcal{M}_{\text{critic}}$
**Ensure:** Generated critique $\Delta y_{\text{unexp}}$ for weak model output
1: $y_{\text{unexp}} \leftarrow \mathcal{M}_{\text{weak}}(q, \mathcal{D})$                   ▷ Generate suboptimal response
2: $y_{\text{exp}} \leftarrow \mathcal{M}_{\text{strong}}(q, \mathcal{D})$                    ▷ Generate preferred response
3: $\mathcal{X}_{\text{pref}} \leftarrow (q, \mathcal{D}, y_{\text{exp}}, y_{\text{unexp}})$         ▷ Construct preference-informed input
4: $\Delta y_{\text{unexp}} \leftarrow \mathcal{M}_{\text{critic}}(\mathcal{X}_{\text{pref}})$            ▷ Generate contrastive critique
5: $\Delta y_{\text{unexp}} \leftarrow \mathcal{G}(\Delta y_{\text{unexp}}, y_{\text{exp}})$      ▷ Refine critique with improvement guidance
6: **return** $\Delta y_{\text{unexp}}$

---

**Algorithm 2** CRITIQUE FINE-TUNING (CFT): Supervised adaptation via synthetic critiques

---

**Require:** Base model $\mathcal{M}_{\text{weak}}$, synthetic dataset $\mathcal{C}$, template $\mathcal{I}_{\text{critic}}$, learning rate $\eta$, epochs $N$
**Ensure:** Critique-aware model $\mathcal{M}_{\text{critic}}$
1: $\mathcal{M}_{\text{critic}} \leftarrow \mathcal{M}_{\text{weak}}$                      ▷ Initialize from weak model
2: **for** epoch $= 1$ to $N$ **do**
3:     **for** each $(q, \mathcal{D}, y_{\text{unexp}}, \Delta y_{\text{unexp}}, y_{\text{exp}}) \in \mathcal{C}$ **do**
4:         $\mathcal{I}_{\text{critic}} \leftarrow (q, \mathcal{D}, y_{\text{unexp}}, y_{\text{exp}})$      ▷ Compose critique context
5:         $\Delta \hat{y}_{\text{unexp}} \sim p_\theta(\cdot \mid \mathcal{I}_{\text{critic}})$            ▷ Predict critique
6:         $\mathcal{L}_{\text{CFT}} \leftarrow -\log p_\theta(\Delta y_{\text{unexp}} \mid \mathcal{I}_{\text{critic}})$     ▷ Compute NLL loss
7:         $\theta \leftarrow \theta - \eta \nabla_\theta \mathcal{L}_{\text{CFT}}$              ▷ Update model
8: **return** $\mathcal{M}_{\text{critic}}$

---

**Algorithm 3** CRITIQUE PREFERENCE OPTIMIZATION (CPO): Alignment via pairwise critique preferences

---

**Require:** Queries $\{q\}$, contexts $\{\mathcal{D}\}$, weak model $\mathcal{M}_{\text{weak}}$, strong model $\mathcal{M}_{\text{strong}}$, initial model $\mathcal{M}_{\text{critic}}$, temperature $\beta$
**Ensure:** Preference-aligned critique model $\mathcal{M}_{\text{critic}}$
1: **for** each $(q, \mathcal{D})$ **do**
2:     $\Delta y_{\text{unexp}}^- \leftarrow \mathcal{M}_{\text{weak}}(q, \mathcal{D})$            ▷ Infer less-preferred critique
3:     $\Delta y_{\text{unexp}}^+ \leftarrow \mathcal{M}_{\text{strong}}(q, \mathcal{D})$           ▷ Infer preferred critique
4:     $\mathcal{P} \leftarrow (\Delta y_{\text{unexp}}^-, \Delta y_{\text{unexp}}^+)$            ▷ Construct preference pair
5: **for** epoch $= 1$ to $N$ **do**
6:     **for** each $\mathcal{P} = (\Delta y^-, \Delta y^+)$ **do**
7:         Compute preference loss $\mathcal{L}_{\text{DPO}}$      ▷ Direct Preference Optimization loss
8:         $\theta \leftarrow \theta - \eta \nabla_\theta \mathcal{L}_{\text{DPO}}$            ▷ Update parameters
9: **return** $\mathcal{M}_{\text{critic}}$

---

**Algorithm 4** CRITIQUE-DRIVEN ALIGNMENT (CDA): Iterative refinement via model-generated critique signals

---

**Require:** Query $q$, document set $\mathcal{D} = \{d_1, \ldots, d_n\}$, generation model $\mathcal{M}_{\text{gen}}$, critique model $\mathcal{M}_{\text{critic}}$, iterations $T$
**Ensure:** Refined, critique-aligned response $y_{\text{exp}}$
1: $y_0 \leftarrow \mathcal{M}_{\text{gen}}(q, \mathcal{D})$               ▷ Initial retrieval-augmented generation
2: **for** $t = 0$ to $T-1$ **do**
3:     $\Delta y_t \leftarrow \mathcal{M}_{\text{critic}}(y_t, q, \mathcal{D})$            ▷ Critique current response
4:     $y_{t+1} \leftarrow \mathcal{M}_{\text{gen}}(y_t \oplus \Delta y_t, q, \mathcal{D})$       ▷ Refine using critique
5: $y_{\text{exp}} \leftarrow y_T$                     ▷ Final critique-aware output
6: **return** $y_{\text{exp}}$

---

## A.6 Additional Experiment Results

In this section, we present additional experimental results to provide a comprehensive understanding of the proposed method and its performance under various conditions.

**Generalization to Out-of-Distribution Data.** To supplement the OOD generalization results in Figure 3, Table 6 provides a complete breakdown of ID and OOD performance across benchmarks and backbones. While the main text reports average performance drops between ID and OOD settings, the detailed analysis reveals that ALIGNRAG reduces the OOD drop significantly (e.g., from 40.3 to 32.2 on Qwen2.5-7B and from 44.0 to 33.1 on LLaMA3.1-8B) compared to SELF-REFINE. Additionally, ALIGNRAG achieves substantial absolute gains on OOD datasets (e.g., +12.4 on HotpotQA and +11.5 on SQuAD for Qwen2.5-7B), demonstrating improved generalization capabilities under domain shifts. These results confirm that the proposed CDA-based alignment strategy enhances model robustness across distributions without overfitting to the training data.

**Robustness under Retrieval Quality Variance.** To evaluate robustness under varying retrieval conditions, we compare Vanilla RAG, SELF-REFINE, and ALIGNRAG in two retrieval scenarios: (*Informative*) and (*Noisy*). Figure 9 summarizes the results. In the *Noisy* scenario, where noisy or misleading retrieval often causes reasoning misalignment, ALIGNRAG consistently outperforms the baselines. For example, on NaturalQuestions (e.g., +5.6 on Qwen2.5-7B) and 2WikiMultiHopQA (e.g., +3.9 on LLaMA3.1-8B)—two tasks particularly sensitive to retrieval quality—ALIGNRAG achieves the largest margins over SELF-REFINE. Even in the *Informative* scenario, where retrieved documents are highly relevant, ALIGNRAG demonstrates superior accuracy (e.g., +1.9 on ASQA and +4.7 on 2WikiMultiHopQA using Qwen2.5-14B). These results illustrate that ALIGNRAG enhances reasoning robustness across a wide range of retrieval quality levels.

**Integration into Existing RAG Pipelines.** To assess the plug-and-play compatibility of our alignment strategy, we integrate it into the INSTRUCTRAG framework across three backbones and evaluate its performance on seven benchmarks. The detailed results, provided in Figure 8, reveal that our alignment approach consistently improves accuracy, both for in-domain datasets (e.g., PopQA, TriviaQA) and OOD datasets (e.g., SQuAD, HotpotQA). Notably, the improvements are particularly pronounced on challenging datasets such as SQuAD (+10.2 on Qwen2.5-14B) and HotpotQA (+8.6 on Qwen2.5-14B) when leveraging the 72B model in our CCS pipeline. These results demonstrate that our method can be seamlessly incorporated into existing RAG pipelines, enabling substantial test-time improvements without requiring modifications to the model architecture or training objectives.

**Superior Alignment Guidance Surpassing Strong LLM Baselines.** To validate the effectiveness of contrastive critique supervision when guided by a 72B model, we report full results across seven QA benchmarks in Table 3. Our method consistently outperforms both vanilla critique synthesis and direct CLM supervision. Notably, the improvements are more pronounced on complex datasets such as HotpotQA and SQuAD, where our method yields gains of up to +9.2 and +6.0 on Qwen2.5-7B, respectively, over direct supervision. These results confirm that contrastive critique signals distilled from a stronger model can significantly enhance the generalization ability of smaller CLMs.

**Different Training Strategies for CLM.** To compare training strategies for the CLM, Table 5 evaluates our proposed Critique Fine-tuning (CFT) approach against Critique Preference Optimization (CPO) A.4. Across three backbones and seven benchmarks, CFT consistently outperforms CPO, particularly on retrieval-sensitive and OOD-heavy tasks such as HotpotQA and SQuAD. For example, on Qwen2.5-14B, CFT raises the average accuracy from 51.0 to 53.6 and improves performance on SQuAD from 20.2 to 26.6. Similarly, on LLaMA3.1-8B, CFT achieves a +4.0 gain in average performance and a +10.0 improvement on SQuAD. These results underscore the superiority of preference-based critique generation over preference-based output generation for CLM training, particularly in retrieval-intensive contexts.

**Ablation on CLM and Contrastive Critique Synthesis.** To supplement the high-level ablation analysis, Table 4 presents detailed results for seven benchmarks under four Critic Language Model (CLM) configurations. While the main text reports averaged scores on seven benchmarks, the table provides detailed results. Introducing a frozen CLM yields noticeable gains over Vanilla RAG (e.g., +3.8 on PopQA and +3.9 on ASQA for Qwen2.5-7B), confirming the utility of auxiliary critique.

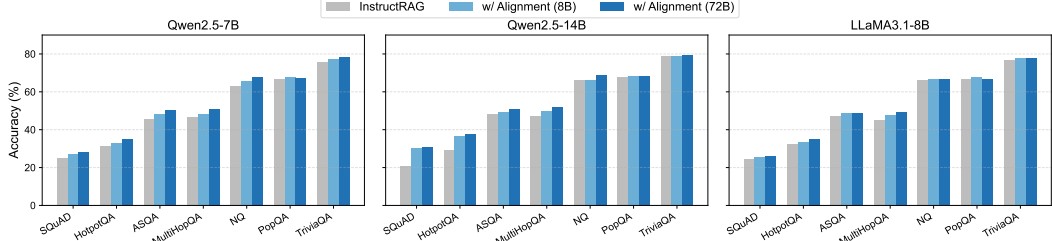

Figure 8: Details of evaluation result of InstructRAG w/o and w/ our Alignment method on three backbones across seven benchmarks.

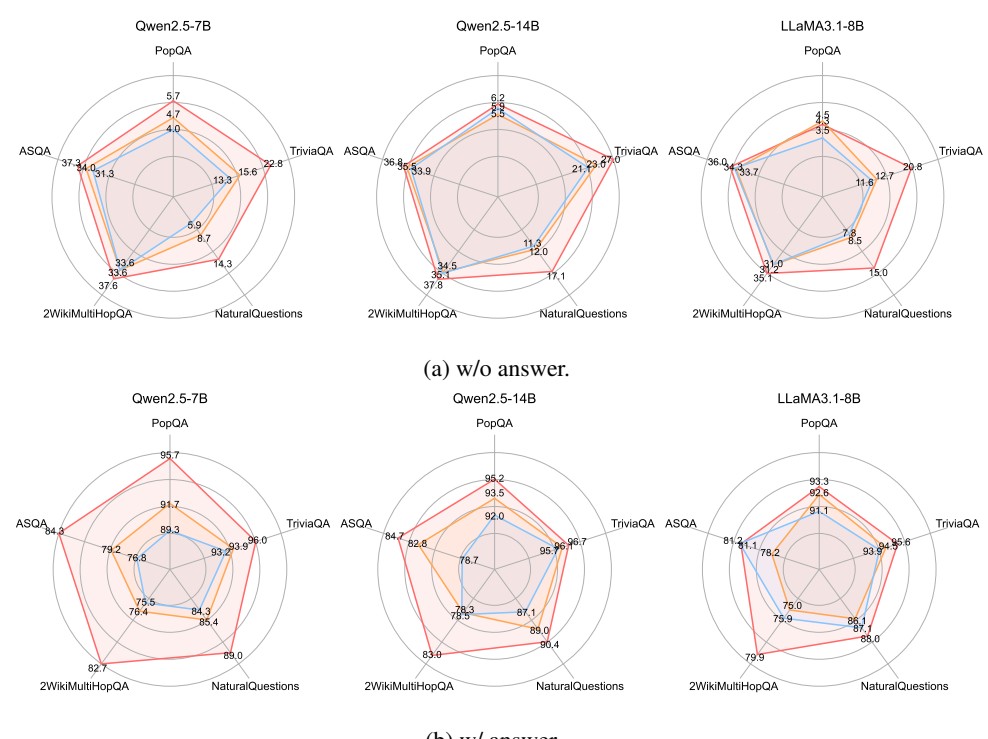

(a) w/o answer.

(b) w/ answer.

Figure 9: Performance of different methods under *Unanswerable* (a) and *Answerable* (b) retrieval conditions. Each radar chart reports the average performance across three instruction-tuned backbones on five benchmarks.

Further training of the CLM amplifies these benefits, particularly for OOD datasets such as SQuAD and HotpotQA. Notably, our contrastive critique synthesis (CCS) achieves the best performance on nearly all benchmarks, including a +2.2 gain on MultiHopQA and +4.7 gain on SQuAD for Qwen2.5-14B. These results demonstrate that contrastive alignment is crucial for generating retrieval-sensitive critiques, leading to consistent and robust improvements across diverse QA scenarios.

Table 3: Detailed results of performance comparison of different trained CLM when using a 72B model to supervise.

| Method | PopQA (acc) | TriviaQA (acc) | NQ (acc) | MultiHopQA (acc) | ASQA (em) | HotpotQA (acc) | SQuAD (acc) | Avg. |
|---|---|---|---|---|---|---|---|---|
| *Vanilla Critique Synthesis* | | | | | | | | |
| Qwen2.5-7B | 66.6 | 76.2 | 65.4 | 46.0 | 47.9 | 25.2 | 17.9 | 49.3 |
| Qwen2.5-14B | 67.5 | 78.9 | 66.8 | 47.5 | 48.7 | 29.6 | 21.2 | 51.5 |
| LLaMA3.1-8B | 66.6 | 75.7 | 64.4 | 43.7 | 47.1 | 22.9 | 13.2 | 47.7 |
| *Direct CLM (72B)* | | | | | | | | |
| Qwen2.5-7B | 66.4 | 78.7 | 66.8 | 46.9 | 49.3 | 24.7 | 18.2 | 50.1 |
| Qwen2.5-14B | 68.6 | 79.3 | 67.8 | 47.4 | 49.3 | 26.8 | 19.7 | 51.3 |
| LLaMA3.1-8B | 67.1 | 78.9 | 66.4 | 45.7 | 49.1 | 24.2 | 15.9 | 49.6 |
| *Ours* | | | | | | | | |
| Qwen2.5-7B | 66.0 | 77.5 | 66.4 | 49.9 | 48.6 | 33.9 | 24.2 | 52.4 |
| Qwen2.5-14B | 66.9 | 79.4 | 68.6 | 50.7 | 49.9 | 35.5 | 25.4 | 53.8 |
| LLaMA3.1-8B | 66.6 | 77.0 | 66.3 | 49.6 | 48.2 | 32.0 | 20.7 | 51.5 |

Table 4: Details of ablation study on the CLM and CCS. Frozen CLM refers to a vanilla LLaMA3.1-8B used as the critic. CCS refers to our proposed contrastive critique synthesis.

| Method | PopQA (acc) | TriviaQA (acc) | NQ (acc) | MultiHopQA (acc) | ASQA (em) | HotpotQA (acc) | SQuAD (acc) | Avg. |
|---|---|---|---|---|---|---|---|---|
| *Vanilla RAG* | | | | | | | | |
| Qwen2.5-7B | 63.7 | 73.2 | 60.2 | 44.7 | 42.8 | 18.5 | 9.0 | 44.6 |
| Qwen2.5-14B | 65.3 | 77.0 | 63.6 | 44.8 | 45.2 | 23.3 | 12.6 | 47.4 |
| LLaMA3.1-8B | 65.0 | 73.4 | 62.0 | 43.0 | 45.2 | 17.1 | 6.1 | 44.5 |
| *w/ Frozen CLM* | | | | | | | | |
| Qwen2.5-7B | 67.5 | 75.1 | 62.5 | 45.4 | 46.7 | 20.6 | 13.4 | 47.3 |
| Qwen2.5-14B | 68.0 | 78.0 | 65.1 | 46.6 | 48.1 | 25.3 | 15.6 | 49.5 |
| LLaMA3.1-8B | 66.1 | 74.1 | 61.4 | 42.8 | 44.7 | 18.8 | 8.7 | 45.2 |
| *w/ Trained CLM* | | | | | | | | |
| Qwen2.5-7B | 66.9 | 76.1 | 64.0 | 46.7 | 46.5 | 26.8 | 19.9 | 49.6 |
| Qwen2.5-14B | 67.1 | 78.4 | 65.7 | 47.6 | 48.8 | 30.0 | 21.9 | 51.4 |
| LLaMA3.1-8B | 65.1 | 74.3 | 62.7 | 42.8 | 46.5 | 25.2 | 16.3 | 47.6 |
| *w/ Trained CLM, w/ CCS (Ours)* | | | | | | | | |
| Qwen2.5-7B | 68.4 | 77.8 | 65.9 | 49.5 | 48.9 | 33.7 | 26.1 | 52.9 |
| Qwen2.5-14B | 68.4 | 79.5 | 67.7 | 49.8 | 48.6 | 34.8 | 26.6 | 53.6 |
| LLaMA3.1-8B | 66.5 | 77.0 | 65.3 | 47.0 | 47.1 | 32.2 | 22.8 | 51.1 |

Table 5: Overall performance comparison of Critic Language Model using different training methods.

| Method | PopQA (acc) | TriviaQA (acc) | NQ (acc) | MultiHopQA (acc) | ASQA (em) | HotpotQA (acc) | SQuAD (acc) | Avg. |
|---|---|---|---|---|---|---|---|---|
| *Qwen2.5-7B* | | | | | | | | |
| CPO | 66.1 | 76.3 | 63.5 | 46.3 | 47.1 | 25.6 | 17.3 | 48.9 |
| Ours | 68.4 | 77.8 | 65.9 | 49.5 | 48.9 | 33.7 | 26.1 | 52.9 |
| *Qwen2.5-14B* | | | | | | | | |
| CPO | 67.5 | 78.6 | 66.1 | 47.4 | 47.7 | 29.5 | 20.2 | 51.0 |
| Ours | 68.4 | 79.5 | 67.7 | 49.8 | 48.6 | 34.8 | 26.6 | 53.6 |
| *LLaMA3.1-8B* | | | | | | | | |
| CPO | 66.4 | 75.0 | 62.7 | 44.0 | 45.7 | 23.2 | 12.8 | 47.1 |
| Ours | 66.5 | 77.0 | 65.3 | 47.0 | 47.1 | 32.2 | 22.8 | 51.1 |

Table 6: Drop in average Out-of-Distribution performance compared to average In-Domain performance across three instruction-tuned backbones. Lower values indicate better generalization capability.

| Method | PopQA (acc) | TriviaQA (acc) | NQ (acc) | MultiHopQA (acc) | ASQA (em) | Avg. | HotpotQA (acc) | SQuAD (acc) | Avg. | Drop. |
|---|---|---|---|---|---|---|---|---|---|---|
| *Qwen2.5-7B* | | | | | | | | | | |
| Vanilla RAG | 63.7 | 73.2 | 60.2 | 44.7 | 42.8 | 56.9 | 18.5 | 9.0 | 13.8 | 43.1 |
| Self-Refine | 65.5 | 74.4 | 61.6 | 45.0 | 45.2 | 58.3 | 21.3 | 14.6 | 18.0 | 40.3 |
| AlignRAG | 68.4 | 77.8 | 65.9 | 49.5 | 48.9 | 62.1 | 33.7 | 26.1 | 29.9 | 32.2 |
| *Qwen2.5-14B* | | | | | | | | | | |
| Vanilla RAG | 65.3 | 77.0 | 63.6 | 44.8 | 45.2 | 59.2 | 23.3 | 12.6 | 18.0 | 41.2 |
| Self-Refine | 67.0 | 78.0 | 65.1 | 46.1 | 47.3 | 60.7 | 24.4 | 16.0 | 20.2 | 40.5 |
| AlignRAG | 68.4 | 79.5 | 67.7 | 49.8 | 48.6 | 62.8 | 34.8 | 26.6 | 30.7 | 32.1 |
| *LLaMA3.1-8B* | | | | | | | | | | |
| Vanilla RAG | 65.0 | 73.4 | 62.0 | 43.0 | 45.2 | 57.7 | 17.1 | 6.1 | 11.6 | 46.1 |
| Self-Refine | 66.1 | 74.1 | 61.4 | 42.8 | 44.7 | 57.8 | 18.8 | 8.7 | 13.8 | 44.0 |
| AlignRAG | 66.5 | 77.0 | 65.3 | 47.0 | 47.1 | 60.6 | 32.2 | 22.8 | 27.5 | 33.1 |

## A.7   Prompt Templates

**Critique Synthesis Prompt.** We propose a novel structured pipeline for generating targeted feedback to train critic models, systematically deriving critiques from contrasting outputs of large language models (LLMs). To ensure the critiques are both consistent and informative, we introduce a *preference-augmented input* as a key component in the critique generation process. This approach is grounded in the use of pairwise comparisons of reasoning paths, which provides two core innovations. First, it constrains the output space of the critique language model (CLM), ensuring consistency and minimizing noise during critique generation [36]. Second, it generates high-quality reasoning traces that facilitate the creation of constructive, fine-grained feedback. The pairwise-path formulation is central to this framework: by contrasting the reasoning processes underlying $y_{unexp}$ (unexpected response) and $y_{exp}$ (expected response), the CLM synthesizes critiques that directly inform model supervision. This is exemplified in Tab. 7 (for rationale generation) and Tab. 9 (for critique generation). This structured methodology not only enhances the quality of the generated critiques but also ensures they are targeted, actionable, and aligned with the requirements of improving weaker models.

**Critique Learning Prompt.** To further advance critique generation, we introduce the concept of *critique learning*, where the objective is to generate a critique, denoted as $\Delta y_{unexp}$, that captures the divergence between expected and unexpected responses while incorporating user-defined preferences. As part of this framework, we present a novel Critique Fine-Tuning (CFT) prompt (see Tab. 10 for details) designed to optimize the learning process for critique generation. Additionally, we explore an alternative training strategy, *Critique Preference Optimization (CPO)*, which explicitly aligns critique generation with user-defined preference signals (see Tab. 11 for the corresponding prompt). These prompts, tailored for critique learning, establish a principled mechanism for training models to generate preference-aligned critiques.

**Critique-driven Alignment Prompt.** We introduce a novel framework, *Critique-driven Alignment (CDA)*, to address reasoning misalignment in retrieval-augmented generation (RAG) systems. CDA reimagines the RAG inference process as a discrete-time dynamical system operating over a latent reasoning space $\mathcal{Y}$. Within this framework, the inference process is iteratively refined by a meta-reasoning module $\mathcal{M}_{critic}$, which critiques intermediate outputs and proposes targeted improvements. This iterative refinement produces a sequence of progressively improved responses, ensuring reasoning alignment.

CDA leverages three distinct prompt types to structure the refinement pipeline effectively:

- **Rationale Generation:** Using the rationale generation template (see Tab. 12), the system generates an initial explanation or reasoning chain to support the initial response $y_0$. This rationale serves as the foundation for critique generation in subsequent steps.

- **Critique Generation:** Using the critique generation template (see Tab. 13), the meta-reasoning module $\mathcal{M}_{critic}$ identifies reasoning gaps or inconsistencies in the intermediate response $y_t$ based on the rationale and provides an actionable critique $\Delta y_t$.

- **Refinement Generation:** Using the refinement generation template (see Tab. 14), the system incorporates the critique $\Delta y_t$ into the generation process to produce the refined response $y_{t+1}$. This ensures that the updated response aligns with the critique feedback while maintaining coherence and relevance to the original query $q$.

By iteratively applying these three prompts, the CDA framework introduces a systematic and controlled refinement process that enhances reasoning alignment and response quality over successive iterations. This novel paradigm ensures that critiques are not only actionable but also effectively integrated into the refinement process to achieve consistent improvements in reasoning accuracy.

Table 7: Rationale generation prompt template for critique synthesis [13].

| Rationale Generation for Critique Synthesis |
| --- |
| **Input:** Read the following documents relevant to the given question: {question} |
| Document [1] (Title: · · · ): {contents} 
 · · · 
 Please identify documents that are useful to answer the given question: "{question}", and explain how the contents lead to the answer: {answer}. |
| If none of the documents is aligned with the answer, in that case, you have to explain the answer only based on your own knowledge, without referring to the provided information. |
| **{task-specific instruction}** |
| **Output:** {rationale} |

Table 8: Task-specific instruction used in rationale generation prompt [13].

| Task-specific Instruction for Rationale Generation |
| --- |
| **ASQA:** Note that the question may be ambiguous and have multiple correct answers. Make sure your response includes all correct answers and provides clear reasoning details followed by a concise conclusion. |
| **PopQA:** Note that the question mainly asks about the object entity that holds a certain relationship with the given subject entity. There may be multiple correct answers. Make sure your response includes all correct answers and provides clear reasoning details followed by a concise conclusion. |
| **TriviaQA / Natural Questions / 2WikiMultiHopQA:** Note that the question may be compositional and require intermediate analysis to deduce the final answer. Make sure your response is grounded and provides clear reasoning details followed by a concise conclusion. |

Table 9: Critique generation prompt template for critique synthesis.

| Critique Generation for Critique Synthesis |
| --- |
| **Input:** Read the following documents relevant to the given question: {question} |
| Document [1] (Title: · · · ): {contents} 
 · · · |
| Here is the given weak rationale: {weak_rationale}. 
 Here is the given gold rationale: {gold_rationale}. |
| First, explain how the gold rationale leads to the answer step by step. |
| Then, identify the errors and hallucinations of the weak rationale, and give constructive criticism for improving the weak rationale to be more aligned with the gold rationale. |
| **Output:** {critique} |

Table 10: Augmented critique generation prompt template for critique fine-tuning (CFT).

| Augmented Critique Generation for Critique Fine-tuning (CFT) |
|---|
| **Input:** Read the following documents relevant to the given question: {question}

Document [1] (Title: · · · ): {contents}
· · ·

Here is the given weak rationale: {weak_rationale}.

Please identify the weaknesses and hallucinations of the rationale, and give constructive criticism for improving the weak rationale.

**Output:**

The critique for the rationale is: {critique}.
The better rationale should be: {gold_rationale}. |

Table 11: Critique generation prompt template for critique preference optimization (CPO).

| Augmented Critique Generation for Critique Preference optimization (CPO) |
|---|
| **Input:** Read the following documents relevant to the given question: {question}

Document [1] (Title: · · · ): {contents}
· · ·

Here is the given weak rationale: {weak_rationale}.

Please identify the weaknesses and hallucinations of the rationale, and give constructive criticism for improving the weak rationale.

**Chosen:** The critique for the rationale is: {weak_critique}.
**Rejected:** The critique for the rationale is: {gold_critique}. |

Table 12: Rationale generation prompt template for Critique-driven Alignment.

| Rationale Generation for Critique-driven Alignment |
|---|
| **Input:** Read the following documents relevant to the given question: {question}

Document [1] (Title: · · · ): {contents}
· · ·
Please identify documents that are useful to answer the given question: "{question}", and explain how the contents lead to the answer: {answer}.

**Output:** {rationale} |

## A.8 Case Study

To provide a concrete illustration of the reasoning misalignment issues our framework addresses, we present a series of case studies. These examples demonstrate how failures can occur at each of the three distinct phases of retrieval-augmented reasoning—Relevance Assessment, Query-Evidence

Table 13: Critique generation prompt template for Critique-driven Alignment.

| Critique Generation for Critique-driven Alignment |
| --- |
| **Input:** Read the following documents relevant to the given question: {question}

Document [1] (Title: · · · ): {contents}
· · ·

Here is the given weak rationale: {weak_rationale}.

Please identify the weaknesses and hallucinations of the rationale, and give constructive criticism for improving the weak rationale.

**Output:** {critique} |

Table 14: Refinement generation prompt template for Critique-driven Alignment.

| Refinement Generation for Critique-driven Alignment |
| --- |
| **Input:** Read the following documents relevant to the given question: {question}

Document [1] (Title: · · · ): {contents}
· · ·

Here is the given weak rationale: {weak_rationale}.
Here is the given critique: critique.

Please correct the weak rationale based on the critique, and write a better rationale to explain how the contents lead to the answer.

**Output:** {refinement} |

Mapping, and Evidence-Integrated Synthesis—even when the initial retrieval is successful. We also include a failure analysis of our own model, ALIGNRAG, to highlight its limitations.

### A.8.1 Illustrating Reasoning Misalignment

**Case 1: Misalignment in Relevance Assessment.** This case demonstrates a failure in the initial reasoning phase, where the model incorrectly dismisses highly relevant evidence. The retriever successfully finds a document containing the correct answer, but the generator's internal relevance assessment fails, causing it to discard the evidence and claim the information is unavailable. This highlights that successful retrieval is insufficient if the model cannot recognize the value of the retrieved content.

Table 15: An example of misalignment in Phase 1 (Relevance Assessment).

| Failure Mode: Relevance Misjudgment | |
| --- | --- |
| **Question** | Who is the mother of Mary in Islam? |
| **Golden Answer** | Hannah |
| **Retrieved Evidence** | Document 2 explicitly states, "Hannah, the mother of Mary..." |
| **Initial Response** | "The given documents are not relevant to the question." |
| **Critic Evaluation** | [Bad] — Model failed to recognize clearly relevant evidence. |
| **Refined Response** | "Hannah is mentioned as the mother of Mary in the provided context." |

**Case 2: Misalignment in Query-Evidence Mapping.** In this scenario, the model fails during the mapping phase. Although multiple documents are retrieved, the model latches onto a document with a vague thematic association (Document 1) while ignoring another document (Document 5) that contains the precise, explicit answer. This mapping failure demonstrates the challenge of aligning the specific query with the most salient evidence span, a critical step for accurate, grounded generation.

Table 16: An example of misalignment in Phase 2 (Query-Evidence Mapping).

| **Failure Mode: Evidence Mapping Failure** | |
| --- | --- |
| **Question** | What name is given to a very long forward pass in football made in desperation? |
| **Golden Answer** | Hail Mary |
| **Retrieved Evidence** | Document 5 directly defines the "Hail Mary pass" in this context. |
| **Initial Response** | The model cites Document 1, which discusses related football terms but ignores the direct definition in Document 5. |
| **Critic Evaluation** | [Bad] — Model missed the most direct and salient piece of evidence. |
| **Refined Response** | Correctly identifies "Hail Mary" using the evidence from Document 5. |

**Case 3: Misalignment in Synthesis.** This case illustrates a failure at the final synthesis stage. The model correctly identifies and internally processes the relevant document containing the answer. However, it fails to integrate this crucial piece of information into its final generated output. The evidence is understood but ultimately omitted, rendering the response incomplete and unhelpful. This shows that even with perfect retrieval and mapping, the synthesis process itself can be a point of failure.

Table 17: An example of misalignment in Phase 3 (Synthesis).

| **Failure Mode: Synthesis Error** | |
| --- | --- |
| **Question** | Who was the director of Alexander? |
| **Golden Answer** | Oliver Stone |
| **Retrieved Evidence** | Document 5 states, "It was directed by Oliver Stone..." |
| **Initial Response** | The model discusses the movie but fails to state the director's name, despite having access to the information. |
| **Critic Evaluation** | [Bad] — Information was present in the evidence but was omitted in the final output. |
| **Refined Response** | "The only director identified in the provided documents is Oliver Stone." |

### A.8.2 Failure Analysis of ALIGNRAG

While ALIGNRAG is designed to correct the misalignments above, it is not without its own failure modes. The most critical weakness arises when the initial retrieval is incomplete or fails to provide any relevant evidence. In such cases, the Critic Language Model (CLM) may attempt to "over-correct" the initial response by injecting factual knowledge from its own parameters, which is not grounded in the provided context. This can lead to factual drift and produce a refined response that is still incorrect but for a different reason.

### A.9 Broader Impact

The AlignRAG framework promises positive societal impact by enhancing the factual reliability and evidence-grounded reasoning of LLMs, potentially leading to more trustworthy AI systems in areas like education, research, and complex decision-support, thereby reducing the spread of unsupported or misaligned information. This can empower users with more accurate and verifiable information. However, challenges and risks must be acknowledged: an increased perception of reliability, even if improved, could lead to over-reliance by users and a reduction in critical scrutiny. Furthermore, the critique mechanism itself, while aiming for better alignment, might inadvertently

Table 18: A representative failure case for ALIGNRAG.

| **Failure Mode: Over-Correction due to Weak Retrieval** | |
|---|---|
| **Question** | When was the first 10 dollar bill made? |
| **Golden Answer** | 1861 |
| **Retrieved Evidence** | No retrieved document explicitly mentioned the year 1861. |
| **Initial Response** | "The first 10 dollar bill was issued in 1911." (Incorrect) |
| **Critic Evaluation** | [Bad] — The critique proposed the year 1914, likely based on the CLM's internal knowledge rather than the provided evidence. |
| **Refined Response** | "The first $10 bill was made in 1914." (Incorrect) |

absorb or amplify subtle biases present in the data used for training the Critic Language Model or in the "expert" examples used for contrastive synthesis if not meticulously curated and audited. Therefore, the responsible development and deployment of such advanced RAG systems necessitate ongoing research into robust bias detection and mitigation techniques, ensuring diversity in training data and retrieved evidence, and promoting digital literacy to encourage critical user engagement with AI-generated content.

