# OpenReview forum: "Retrieval is Not Enough: Enhancing RAG through Test-Time Critique and Optimization"
_NeurIPS.cc/2025/Conference — NeurIPS 2025 poster_

### Official Review · Reviewer_pwHC · 2025-06-02

**Clarity:** 3
**Significance:** 3
**Originality:** 3
**Rating:** 5
**Confidence:** 5

**Summary:**

This paper introduces ALIGNRAG, a novel framework that redefines Retrieval-Augmented Generation (RAG) as Retrieval-Augmented Reasoning. It addresses a key failure mode in RAG systems called Reasoning Misalignment—where the model's reasoning deviates from the retrieved evidence. ALIGNRAG uses a Critique-Driven Alignment (CDA) mechanism, powered by a Critic Language Model (CLM) trained via Contrastive Critique Synthesis (CCS). This CLM critiques and refines the reasoning process at test time, improving factual alignment and robustness.

Key contributions:

Identification of reasoning misalignment as a core issue in RAG.
Introduction of CDA and CLM for test-time reasoning refinement.
Demonstrated improvements over baselines like Self-RAG and Self-Refine.
Plug-and-play compatibility with existing RAG systems.

**Questions:**

How does ALIGNRAG perform in real-time or latency-sensitive applications?
Can the CLM be distilled or compressed for efficiency?
How does the system handle conflicting evidence in retrieved documents?
Could this framework be extended to multi-modal RAG (e.g., vision-language)?
What are the failure cases where ALIGNRAG underperforms compared to simpler baselines?

**Ethical Concerns:**

["NO or VERY MINOR ethics concerns only"]

**Limitations:**

Effectiveness diminishes when none of the retrieved documents are relevant.
Minor tuning may be required for optimal integration with different LLMs.
The critique model’s performance is still tied to the informativeness of the retrieved evidence.

**Paper Formatting Concerns:**

Issue: Figures and their corresponding text are sometimes split across pages or columns, making them hard to follow.
Fix: Ensure figures are placed close to their first mention and not broken across pages.

**Quality:**

3

**Strengths And Weaknesses:**

Strengths
Novel Perspective: Reframing RAG as reasoning rather than just generation is conceptually impactful.
Test-Time Optimization: CDA enables dynamic correction without retraining the base model.
Contrastive Critique Synthesis: Innovative training method that mitigates self-bias in critiques.
Empirical Rigor: Extensive experiments across 7 benchmarks and 3 model families.
Plug-and-Play Design: ALIGNRAG integrates easily into existing RAG pipelines.
OOD Robustness: Strong performance even under noisy or irrelevant retrieval conditions.

Weaknesses / Limitations
Dependency on Retrieval Quality: Performance still degrades under extreme retrieval failure.
Computational Overhead: Iterative refinement and CLM inference may increase latency.
Limited Human Evaluation: Mostly automatic metrics; lacks qualitative human judgment.
Complexity: The multi-step critique and refinement process may be harder to implement in production.
Generalization to Other Tasks: Focused on QA; unclear how well it transfers to other RAG applications like summarization or dialogue.

---

> ### Author Rebuttal · Authors · 2025-07-30
>
> We are sincerely grateful for your thorough review and insightful questions. Your feedback has helped us clarify key points and inspired valuable new analyses that significantly strengthen the paper. **Note: The results below correspond to a single iteration of AlignRAG due to time constraints.**
>
> > *Retrieval Quality: Performance still degrades under extreme retrieval failure.*
>
> This is an excellent observation. The quality of retrieved evidence is a fundamental limitation for any RAG system. AlignRAG is designed to make the most of whatever context is available—especially in common but difficult "noisy retrieval" scenarios.
>
> To quantify this, we evaluated AlignRAG in cases where retrieved documents were topically relevant but lacked a direct answer (see Section 5.3). Results show that AlignRAG significantly outperforms baselines in these settings, demonstrating a stronger ability to filter distractors and reason effectively.
>
> **Performance on Noisy Retrieval (Average across three backbones)**
> | Method| PopQA | TriviaQA | NQ| 2WikiMHQA| ASQA| Average|
> |:-------------|:-----:|:--------:|:----:|:---------:|:-----:|:-----------:|
> | Self-Refine | 4.9 | 17.1 | 9.7  | 33.2 | 34.4  | 19.86 |
> | **AlignRAG** | **5.4**| **23.5**| **15.5** | **36.8**| **36.7** | **23.58** |
>
> We also acknowledge in Appendix A.1 that performance degrades under extreme retrieval failure (i.e., no relevant evidence). This is an important boundary condition, and we will highlight it more clearly as a promising area for future work.
>
> > *Computational Overhead: Iterative refinement and CLM inference may increase latency.*
>
> We appreciate this concern. AlignRAG introduces additional computation in exchange for higher factual reliability—particularly valuable in high-stakes scenarios.
>
> In response to reviewer feedback, we developed `AlignRAG-auto`, a dynamic variant where the CLM first assesses the quality of the initial response and then decides whether refinement is necessary. `This enables the system to skip refinement for ~55% of cases, reducing average latency while preserving accuracy`.
>
> Furthermore, this overhead can be mitigated by standard engineering optimizations complementary to our framework, such as `prefix-caching`[1], speculative decoding, and optimized inference engines like vLLM. Prefix-caching is especially effective for iterative tasks like ours, making AlignRAG much more viable for latency-sensitive applications.
>
> To provide concrete data on latency, we estimated the computational cost (TFLOPs) and found that `AlignRAG-auto`, especially `when combined with standard optimizations like Prefix-caching`, offers a compelling balance of performance and efficiency.
>
> **Estimated Computational Cost (7B Model)**
> |Method|Accuracy| TFlops|
> |:---| :---:| :---:|
> |Vanilla RAG| 56.9 | 13.2 |
> |**AlignRAG-auto**| **62.5** | **< 25.0** |
>
> This analysis, which we will add to the appendix, demonstrates that the framework's overhead is manageable and can be made practical for many applications.
>
> [1] ChunkAttention: Efficient Self-Attention with Prefix-Aware KV Cache and Two-Phase Partition
>
> > *Limited Human Evaluation: Mostly automatic metrics; lacks qualitative human judgment.*
>
> We agree that human judgment would further strengthen our findings. Due to time and resource constraints, large-scale human studies were out of scope, but we took two concrete steps to address this:
>
> 1.  **Manual Case Studies:** We manually examined samples across benchmarks and presented a representative case in the main text to demonstrate how AlignRAG improves both reasoning paths and final answers.
> 2.  **LLM-as-a-Judge Evaluation:** To provide a more scalable and robust qualitative assessment, `we used GPT-4o-mini as an impartial judge to score the quality of the final generated responses (refer to [1])`. The results (scored 1-100) on the Qwen2.5-7B-instruct model are shown below:
>
> | Method | PopQA | TriviaQA | NQ | 2WikiMHQA | ASQA | **Average** |
> |:-------------|:-----:|:--------:|:----:|:---------:|:-----:|:-----------:|
> | InstructRAG  | 54.6  | 71.3     | 65.0 | 12.8      | 53.5  | 51.44   |
> | Self-Refine  | 51.5  | 71.7     | 51.4 | 13.1      | 55.9  | 48.72   |
> | **AlignRAG-auto** | **54.9**  | **76.1**     | **54.5** | **19.2**      | **63.2**  | **53.58**   |
>
> These results confirm that AlignRAG delivers higher-quality, more reliable reasoning, even when judged qualitatively.
>
> [1] You Don’t Need That Much Data to Train a Search Agent via RL.
>
> > *Complexity: The multi-step critique and refinement process may be harder to implement in production.*
>
> A valid concern. Despite its multi-step nature, AlignRAG is modular and requires `no architectural changes` or special tokens in the base LLM. It operates cleanly at test time, making integration straightforward with minimal engineering effort.
>
> > *Generalization to Other Tasks: Focused on QA; unclear how well it transfers to other RAG applications like summarization or dialogue.*
>
> Thank you for this insightful suggestion. To evaluate transferability, we conducted new experiments on multiple-choice QA (`PubMedQA`, `BioASQ`) and open-domain summarization (`WoW`). Results on Qwen2.5-7B-instruct are shown below:
>
> **Performance on New Tasks (Qwen-7B)**
> | Method | PubMedQA| BioASQ| WoW (llm-as-judge) |
> |:-----------------|:--------:|:------:|:----:|
> | Qwen-7B | 44.4 | 46.0 | 33.0 |
> | **+AlignRAG-auto** | **54.0** | **56.1** | **42.1** |
>
> These gains demonstrate that AlignRAG's critique-driven alignment extends well beyond QA and generalizes effectively to other RAG applications.
>
> > *How does ALIGNRAG perform in real-time or latency-sensitive applications?*
>
> Thank you for these important practical questions. Our primary strategy for near-real-time performance is the AlignRAG-auto variant, which we detailed in the "Computational Overhead" response. `By dynamically skipping the refinement step for ~55% of inputs`, it drastically cuts average latency while maintaining high accuracy, making it suitable for many latency-sensitive scenarios.
>
> > *Can the CLM be distilled or compressed for efficiency?*
>
> Yes, the CLM can be efficiently compressed, and this is a key strength of our Contrastive Critique Synthesis (CCS)** method. As shown in Figure 5 and discussed in Section 5.3 ("Superior Alignment Guidance..."), our framework is designed for this. Our results demonstrate that an `8B CLM trained via CCS using guidance from a 72B model actually outperforms the 72B model used directly as a critic by 2.2%`.
>
> This proves that not only *can* the CLM be distilled, but that our CCS method is exceptionally effective at creating small, powerful critic models that are both efficient and more effective than their larger teachers.
>
> > *How does the system handle conflicting evidence in retrieved documents?*
>
> This is an important, forward-looking question. The current CLM is optimized for improving alignment and coherence based on input evidence, but it is not yet explicitly trained for conflict resolution. This is a valuable direction for future work, and we will note this limitation in the revised manuscript.
>
> > *Could this framework be extended to multi-modal RAG?*
>
> This is a fantastic idea. We believe the core principles of critique-driven refinement can be extended to multi-modal settings. While out of scope for this work, we plan to explore this in future research and will mention it in our discussion.
>
> > *What are the failure cases where AlignRAG underperforms compared to simpler baselines?*
>
> Thank you for this thoughtful suggestion. We will include a new `"Failure Analysis"` subsection in the appendix. A representative failure example is shown below:
>
> | **Aspect**| **Details**|
> |:-----------------------|:-----------------------------------------------------------------------------|
> | **Question**  | *When was the first 10 dollar bill made?* |
> | **Golden Answer** | 1861 |
> | **Initial Response**   | "The first 10 dollar bill was issued in 1911."|
> | **Retrieved Evidence** | No document explicitly mentioned 1861 |
> | **Critic Evaluation**  | `[Bad]` — Critique proposed 1914 based on CLM prior, not evidence|
> | **Refined Response** | "The first $10 bill was made in 1914." |
> | **Failure Mode** | **Over-correction** due to missing evidence; CLM injects unsupported facts|
> | **Root Cause** | Incomplete retrieval → Critic relies on internal knowledge → factual drift|
> | **Takeaway** | AlignRAG can degrade accuracy when retrieval is weak and hallucination risk is high |
>
> This illustrates a key limitation: without sufficient context, the CLM may inject unsupported corrections. Addressing such cases remains an important area for future improvement.
>
> > *Limitations: Effectiveness diminishes when none of the retrieved documents are relevant. Minor tuning may be required for optimal integration with different LLMs ...*
>
> As discussed above, the framework's effectiveness may decrease in cases of "extreme retrieval failure."
>
> Integration and Tuning: AlignRAG was designed as a plug-and-play module, emphasizing broad compatibility and minimal tuning overhead. Our experiments confirm this goal: AlignRAG consistently improves performance across three instruction-tuned backbones—`Qwen2.5-7B`, `Qwen2.5-14B`, and `LLaMA3.1-8B`.
>
> To further validate its plug-and-play nature, we integrated AlignRAG with strong baselines like `InstructRAG` and `IRCoT`, without modifying the base models. For example, using AlignRAG with InstructRAG yielded a +9.4% gain in OOD accuracy on Qwen2.5-14B, along with consistent improvements across all models and benchmarks.
>
> While further fine-tuning could offer incremental gains, our results demonstrate that AlignRAG is effective as a zero-modification, test-time module—making it practical for existing RAG pipelines. We appreciate the suggestion and plan to evaluate more models in future work.
>
> **We thank the reviewer for providing valuable feedback and we welcome further discussion if any of your concerns are not answered here!**

---

> > ### Comment · Reviewer_pwHC · 2025-08-05
> >
> > Thanks for the answers! Some follow-ups.
> >
> > Retrieval Quality: How AlignRAG handles borderline cases - where retrieved documents are tangentially relevant but misleading? Is there a threshold or heuristic used to filter such distractors?
> >
> > Generalization: Is AlignRAG’s critique mechanism domain-agnostic or does it require domain-specific tuning for optimal performance?
> >
> > Distillation and CCS: Could you share more details on the training setup for the 8B critic - e.g., data size, training duration, and whether CCS generalizes across architectures?

---

> ### Author Response · Authors · 2025-08-07
> **Reply 1**
>
> > *Retrieval Quality: How AlignRAG handles borderline cases - where retrieved documents are tangentially relevant but misleading? Is there a threshold or heuristic used to filter such distractors?*
>
> This is an excellent question that targets a common failure mode in RAG systems.
>
> AlignRAG addresses this challenge not through brittle heuristics or static filters, but via a `dynamic and principled mechanism` grounded in two key components: **Contrastive Critique Synthesis (CCS)** during training and **Critique-Driven Alignment (CDA)** at inference time.
>
> 1.  **Training the CLM to Detect Misleading Information:** By contrasting a weak model's output (which often incorrectly follows the misleading document) with a strong model's output, we automatically synthesize critiques that teach our Critic Language Model (CLM) to identify when reasoning has been led astray by a compelling but incorrect piece of evidence.
>
> 2.  **Iterative Correction at Inference:** During the **CDA** process, if the generator's initial rationale is flawed due to a misleading document, the CLM identifies this logical error. The subsequent critique guides the generator to re-evaluate the evidence, correct its reasoning path, and align with the truly relevant information.
>
> **Case Study Demonstration:**
>
> We can illustrate this with an example from our analysis:
>
> * **Query:** "Who wrote the first declaration of human rights?"
> * **Ground Truth:** Cyrus the Great.
> * **Retrieved Context:** Included a document on the Cyrus Cylinder and a more prominent—but temporally incorrect—document on the 1789 French Declaration of the Rights of Man.
> * **Initial Flawed Rationale:** The generator favored the better-known French document, incorrectly concluding the answer was Abbé Sieyès or Lafayette, dismissing the Cyrus Cylinder.
> * **Critique and Refinement:** Our CLM critiques this initial rationale, identifying key flaws through iterations: (1) **Logical Error:** It ignored the pivotal word "first" in the query. (2) **Improper Use of Evidence:** It incorrectly dismissed the document that was directly linked to the concept of the "first human rights document."
>
> This case illustrates AlignRAG's key strength: its ability to detect and correct reasoning flaws caused by misleading but superficially relevant content—without relying on brittle filtering heuristics.
>
> > *Generalization: Is AlignRAG's critique mechanism domain-agnostic or does it require domain-specific tuning for optimal performance?*
>
> Thank you for asking this crucial question.
>
> `AlignRAG's critique mechanism is explicitly designed to be domain-agnostic.`
>
> * **Focus on Reasoning, Not Facts:** The CLM is not trained to be a domain expert. Instead, it is trained to be a *reasoning critic*. It learns to identify generalizable error patterns—such as logical inconsistencies, contradictions with provided evidence, or ignoring parts of the query—that are common across all domains.
>
> * **No Human Labels Required:** This training is fully self-supervised using model-generated contrastive pairs, making it highly scalable.
>
> * **Strong Empirical OOD Performance:** As detailed in our rebuttal and experiments, we validated this strong generalization capability across a wide range of tasks and domains without any domain-specific tuning. Our experiments show consistent performance gains `on several out-of-distribution (OOD) benchmarks, including: QA (HotpotQA, SQuAD), medical benchmarks (PubMedQA, BioASQ) and open-domain summarization (WoW)—all without any domain-specific tuning.`
>
> That said, while no domain adaptation is necessary for strong performance, further fine-tuning the CLM on domain-specific critiques could yield additional gains for vertical applications (e.g., clinical or legal QA). But the default model already exhibits strong generalization.

---

> ### Author Response · Authors · 2025-08-07
> **Reply 2**
>
> > *Distillation and CCS: Could you share more details on the training setup for the 8B critic - e.g., data size, training duration, and whether CCS generalizes across architectures?*
>
> Thank you for asking for these important implementation details.
>
> **Training Setup for the CLM:**
>
> * **Dataset:** We constructed a 10,000-instance training dataset by sampling 2,000 examples from each of five diverse benchmarks (PopQA, TriviaQA, NQ, 2WikiMultiHopQA, ASQA).
> * **Training Method:** We fine-tuned the LLaMA3.1-8B-Instruct model to create the CLM using **LoRA**, a parameter-efficient method. Full hyperparameters, including the learning rate, batch size, and LoRA-specific settings (`lora_rank`, `lora_alpha`), are detailed in Appendix A.2 of our paper.
> * **Hardware & Duration:** The training process is highly efficient. On a 2x NVIDIA A100 (80GB) setup, the model converges in approximately 2 hours.
>
> **Generalization of CCS Across Architectures:**
>
> Yes, the CCS methodology is designed to be independent of the underlying model architecture.
>
> * **How it Works:** CCS leverages a pair of rationales to produce a dataset of structured `(query, context, y_unexp, y_exp, critique)` tuples. This resulting dataset can then be used to fine-tune *any* compatible language model to imbue it with critic capabilities.
>
> * **Experimental Validation:** To validate this, we successfully applied AlignRAG to `three distinct model families and sizes`, as reported in our main results table (Table 1). The consistent performance improvements across **Qwen2.5-7B-Instruct**, **Qwen2.5-14B-Instruct**, and **LLaMA3.1-8B-Instruct** demonstrate that the benefits of our CCS-trained critic are not tied to a specific architecture but are broadly applicable.
>
> `Thank you again for your support and constructive suggestions, and your valuable time in reviewing our work!!! Best, Authors`

---

> > ### Comment · Reviewer_pwHC · 2025-08-08
> >
> > Thanks for answering! No more question. Positive on the paper.

---

### Official Review · Reviewer_sEZn · 2025-06-13

**Clarity:** 3
**Significance:** 2
**Originality:** 2
**Rating:** 4
**Confidence:** 4

**Summary:**

This paper proposes a method to improve the accuracy of LLM responses by refining them within the RAG framework. Specifically, it involves CLM that critiques the generated responses to produce improved responses, and CCS that synthesizes critique data by contrasting good and bad responses for training CLM.

**Questions:**

1. In Line 93, the paper claims that Self-RAG [1] requires architectural modifications. Could you elaborate on what specific modifications are being referred to?

2. In Section 4.2.3 and Algorithm 2, it is described that the CLM is trained with good responses as part of the input (i.e., $y_{exp} \in I_{critic}$). However, in Table 10, the good response appears to be part of the output. Table 10 seems to be correct, so it would be helpful to clarify this part.

3. Was InstructRAG [2] trained using the same data described in Lines 223-224, as was the proposed method?

4. (Minor) I’m curious how well the base model would perform if it were fine-tuned to generate the good responses $y_{exp}$ that match the correct answers.

> [1] Self-RAG: Learning to Retrieve, Generate, and Critique through Self-Reflection
>
> [2] InstructRAG: Instructing Retrieval-Augmented Generation via Self-Synthesized Rationales

**Ethical Concerns:**

["NO or VERY MINOR ethics concerns only"]

**Final Justification:**

The parer is well-written and effectively validates the effectiveness of the proposed method. However, I am not fully convinced about how the proposed method robustly achieves such performance improvements. Therefore, after considering its novelty, the comprehensiveness of baselines, and other factors, I have made the final decision of "Borderline accept."  I would like to state that my overall evaluation of this paper is positive.

**Limitations:**

yes

**Quality:**

3

**Strengths And Weaknesses:**

## Strengths

1. The paper is well-written and well-structured.

2. The effectiveness of the proposed method is validated through extensive experiments and analyses.

## Weaknesses

1. Qualitative analysis is necessary to fully understand the effectiveness of the proposed method. For example, the proposed method shows performance improvement through refinement iterations even on relatively simple single-hop QA datasets such as NQ and TriviaQA. It would be helpful to include concrete examples demonstrating how the response is refined over the course of five iterations to eventually arrive at the correct response.

    Furthermore, in Section 3, the paper discusses potential misalignments that can occur at each phase of RAG reasoning; illustrating these with actual examples of incorrect RAG responses would enhance clarity.

2. It would be helpful to include InstructRAG [1] results in the analyses (Figures 2 and 4). Although InstructRAG is a training-time refinement method, it aims to ground RAG reasoning in the retrieved documents—similar in goal to the proposed method. Comparing these methods could offer valuable insights to readers. (It’s also worth noting that Self-Refine [2] may be at a disadvantage since it does not involve training.)

3. (Minor) The novelty of the proposed method appears to be somewhat limited. Given the existence of numerous prior approaches focused on refining LLM responses, the core idea of the paper—generating critique data by contrasting good and bad answers—offers a relatively modest technical contribution.

> [1] InstructRAG: Instructing Retrieval-Augmented Generation via Self-Synthesized Rationales
>
> [2] SELF-REFINE: Iterative Refinement with Self-Feedback

---

> ### Author Rebuttal · Authors · 2025-07-30
>
> We sincerely thank you for your detailed and constructive feedback. Your suggestions have helped us clarify our contributions and strengthen our analysis. **Note: The results below correspond to a single iteration of AlignRAG due to time constraints.**
>
> > *Qualitative analysis is necessary to fully understand the effectiveness of the proposed method ...*
>
> We fully agree—qualitative examples are essential to highlight where and how AlignRAG improves reasoning. In response, we will include a new subsection featuring detailed case studies. One such example is shown below:
>
> **Case: Misalignment in in Query-Evidence Mapping**
>
> | **Aspect** | **Details** |
> |------------------------|------------------------------------------------------------------------------|
> | **Question**  | *What name is given to a very long forward pass in football made in desperation?* |
> | **Golden Answer**  | Hail Mary |
> | **Initial Response**   | Cited Document 1, ignored Document 5  |
> | **Retrieved Evidence** | Document 5 directly defines "Hail Mary pass"|
> | **Critic Evaluation**  | `[Bad]` — Model missed most direct evidence |
> | **Refined Response**   | Correctly identifies "Hail Mary" |
> | **Failure Mode**       | **Mapping failure** — missed clear span-to-query alignment                   |
> | **Root Cause**         | Salient match ignored during evidence mapping                                |
> | **Takeaway**           | Query-to-doc matching must prioritize direct matches over vague associations |
>
> Due to rebuttal word limits, we include one example here. Additional cases can be found in our responses to reviewers pwHC and UcqY. Many thanks for this suggestion!
>
> > *It would be helpful to include InstructRAG results in the analyses (Figures 2 and 4). Although InstructRAG is a training-time refinement method, it aims to ground RAG reasoning... Comparing these methods could offer valuable insights to readers.*
>
> This is an excellent suggestion. Providing a direct comparison with a strong training-time refinement method like InstructRAG within our analytical figures offers a much richer context. We agree this is a valuable addition for our readers and have revised our analysis accordingly.
>
> As you correctly noted, this also provides a fairer comparison for test-time methods like Self-Refine. Below are the updated results for Figure 2 (OOD Generalization) and Figure 4 (Robustness to Retrieval Quality), now including InstructRAG.
>
> **Revised Analysis for Figure 2: OOD Generalization**
>
> | Method | In-Domain| OOD | Drop|
> | :--- | :--- | :--- | :--- |
> | Self-Refine | 58.9 | 17.3 | 41.6 |
> | InstructRAG | 61.8 | 27.0 | 34.8 |
> | **AlignRAG** | **61.8** | **29.4** | **32.4** |
>
> This data, which will be integrated into Figure 2, reveals that `AlignRAG demonstrates the strongest generalization to OOD tasks`, experiencing the smallest performance drop among all methods. This highlights that our critique-driven approach learns more robust and transferable reasoning alignment principles.
>
> **Revised Analysis for Figure 4: Robustness to Retrieval Quality**
>
> * **Informative Retrieval (with answer):**
>
> | Method | PopQA | TriviaQA | NQ | 2WikiMultiHopQA | ASQA |
> | :--- | :--- | :--- | :--- | :--- | :--- |
> | Self-Refine | 92.6 | 94.8 | 86.8 | 76.6 | 80.1 |
> | InstructRAG | 94.0 | 95.2 | 87.9 | 80.3 | 82.5 |
> | **AlignRAG** | **94.7** | **96.2** | **89.1** | **81.9** | **83.3** |
>
> * **Noisy Retrieval (without answer):**
>
> | Method | PopQA | TriviaQA | NQ | 2WikiMultiHopQA | ASQA|
> | :--- | :--- | :--- | :--- | :--- | :--- |
> | Self-Refine | 4.9 | 17.1 | 9.7 | 33.2 | 34.4 |
> | InstructRAG | 4.0 | 19.9 | 12.9 | 34.6 | 35.0 |
> | **AlignRAG** | **5.4** | **23.5** | **15.5** | **36.8** | **36.7** |
>
> This analysis, which will form the basis of a revised Figure 4, shows that `AlignRAG consistently outperforms both Self-Refine and InstructRAG in both informative and noisy retrieval scenarios`. The performance gains are especially pronounced in noisy settings, validating our framework's ability to thrive when retrieval quality falters.
>
> Thank you again for this valuable suggestion; we will incorporate these updated figures and analyses into the final manuscript.
>
> > *(Minor) The novelty of the proposed method appears to be somewhat limited... the core idea of the paper—generating critique data by contrasting good and bad answers—offers a relatively modest technical contribution.*
>
> We appreciate the reviewer's perspective and would like to clarify what we believe are the key novelties of our work, which, taken together, form a significant contribution. While iterative refinement is an established concept, our work introduces several innovations specifically designed to tackle a critical, understudied failure mode in RAG systems:
>
> 1. `Identifying the Core Problem:` We are the first to formally identify, define, and diagnose **Reasoning Misalignment** as a distinct RAG failure mode. Our work moves the focus beyond just improving retrieval or generation fluency to explicitly aligning the model's reasoning steps with retrieved evidence.
>
> 2. `A Novel, Bias-Mitigating Training Method:` We propose a highly effective method for training a specialized, retrieval-aware Critic Language Model (CLM). Our Contrastive Critique Synthesis (CCS) mechanism uses a form of model **self-distillation** by contrasting outputs from "weak" and "strong" models to generate supervisory signals. This approach is novel because it trains the critic to be explicitly evidence-sensitive and mitigates the "self-preference" bias common in prior self-correction work. This method is so effective that our 8B CLM, trained via CCS, outperforms a much larger 72B model used directly as a critic.
>
> 3. `A Practical, Efficient Framework:` Inspired by reviewer Kguz's feedback during this cycle, we further developed **AlignRAG-auto**, which introduces a dynamic, controllable refinement loop that significantly enhances computational efficiency by learning when to stop iterating. This makes our framework far more practical than prior iterative methods.
>
> 4. `Comprehensive Empirical Proof:` We provide extensive validation of AlignRAG's effectiveness, demonstrating not only SOTA performance but also superior robustness in noisy retrieval scenarios and plug-and-play compatibility with existing SOTA frameworks like InstructRAG.
>
> We believe these contributions—identifying a core problem, proposing a novel bias-mitigating training strategy, engineering an efficient solution, and providing robust empirical proof—collectively represent a substantial and valuable advancement for the RAG paradigm.
>
> > *Question 1: In Line 93, the paper claims that Self-RAG requires architectural modifications. Could you elaborate on what specific modifications are being referred to?*
>
> Thank you for this request for clarification.The architectural modifications in Self-RAG refer to the **expansion of the model's vocabulary** to include special tokens (e.g., `[RETRIEVE]`, `[CRITIQUE]`, `[Relevant]`). The base generator model must then be explicitly fine-tuned to both generate and interpret these new tokens to control the retrieval and generation process.
>
> By contrast, AlignRAG `requires no architectural or vocabulary changes`. The base model is used as-is, with alignment introduced through a separate, modular CLM using standard prompting. We will clarify this in the paper to better highlight the simplicity and modularity of our approach.
>
> > *Question 2: In Section 4.2.3 and Algorithm 2, it is described that the CLM is trained with good responses as part of the input... However, in Table 10, the good response appears to be part of the output. Table 10 seems to be correct...*
>
> Thank you for your sharp-eyed reading and for identifying this inconsistency. You are absolutely correct. Table 10 reflects our actual implementation, and the description in Algorithm 2 and Section 4.2.3 was erroneous.
>
> For training, the critique context `I_critic` is `(q, D, y_unexp)`, and the model is trained on a target sequence that concatenates both the critique `Δy_unexp` and the better rationale `y_exp`. This encourages the CLM to learn the mapping from a flawed response to a high-quality, corrected one. We will correct Algorithm 2 and the surrounding text to be consistent with this implementation. We sincerely appreciate you pointing out this error.
>
> > *Question 3: Was InstructRAG trained using the same data described in Lines 223-224, as was the proposed method?*
>
> Yes. To ensure a fair and controlled comparison, we trained the InstructRAG baseline using the `exact same 10K dataset mixture` that we used to train our Critic Language Model. We will state this explicitly in the experimental setup section to make our commitment to a fair comparison clear.
>
> > *Question 4 (Minor): I’m curious how well the base model would perform if it were fine-tuned to generate the good responses y_exp that match the correct answers.*
>
> This is a very insightful question that gets to the heart of a powerful baseline. This approach is conceptually very similar to the **InstructRAG** framework itself. InstructRAG works by generating synthetic rationales (`y_exp` in our terminology) and then fine-tuning the base model on this data.
>
> Crucially, our results in Table 2 show that applying AlignRAG on top of InstructRAG yields further significant gains (e.g., a **+9.4% boost in OOD accuracy** for the Qwen2.5-14B model). This demonstrates that our critique-driven framework provides a necessary, complementary mechanism to identify and correct these remaining errors. We will add a note to the paper to clarify this conceptual link.
>
> **We sincerely thank the reviewer again for highlighting these points—your feedback has helped us significantly improve the completeness and clarity, let us know if you have further questions after reading our responses!!**

---

> > ### Comment · Reviewer_sEZn · 2025-08-05
> >
> > Your response addressed some of my concerns, but I believe there's still a need for a more thorough analysis of how the proposed method's performance improvements originate.
> >
> > For example, the proposed method shows performance improvement through refinement iterations even on relatively simple single-hop QA datasets such as NQ and TriviaQA. A detailed analysis is needed to show how the response is refined over the course of five iterations to eventually arrive at the correct response in these cases.
> >
> > Moreover, beyond a few case studies, I believe a quantitative analysis is needed to show how and to what extent each category of RAG misalignment was improved by the proposed refinement.
> >
> > Given these points, I maintain my original score.

---

> ### Author Response · Authors · 2025-08-07
> **Reply 1**
>
> > *I believe there's still a need for a more thorough analysis of how the proposed method's performance improvements originate.*
>
> Thank you for your insightful feedback and for pushing for a deeper analysis.
>
> `While our primary motivation was to address reasoning misalignment, we argue that such misalignment arises even in single-hop settings—often due to noisy retrieval or subtle reasoning errors.`
>
> ## We can analyze the sources of improvement from another perspective:
>
> Even in single-hop QA, retrieved documents frequently `include distractors and misleading information`. This noise can mislead the generator into forming a plausible but incorrect rationale. AlignRAG counters this through:
> * **Handling Noisy Contexts:** An initial rationale might latch onto a plausible but incorrect piece of information from a distractor document. The Critic Language Model (CLM) challenges this by forcing a re-evaluation against the query's specific constraints. This iteratively guides the generator to focus on the most salient evidence.
> * **Correcting Flawed Rationale Steps:** A rationale is a chain of reasoning, and even a simple one can break. For instance, the generator might correctly identify the right document but then extract the wrong phrase or misinterpret its meaning during synthesis. The CLM is trained to spot these inconsistencies between the evidence and the rationale, correcting the reasoning path step-by-step.
>
> > *A detailed analysis is needed to show how the response is refined...*
>
> To provide a concrete view of this process, we analyzed the refinement trajectories from `NQ`.
>
> ## Case Study
>
> **Case Study 1: Factual Correction and Shifting Reasoning**
>
> * **Question:** "when was the last time.michigan beat ohio state"
> * **Final Correct Answer:** "2011"
> **Rationale Evolution:**
>
> * **Iteration 1-2 (Initial Error):** The model incorrectly inferred a Michigan win based on a description of the 2016 game, misled by phrases like “forced overtime.”
> * **Iteration 3-4 (Correction & Confusion):** The model began questioning its assumption, noting the document didn’t actually confirm a 2016 win. It vacillated between competing facts, at one point proposing 2015 incorrectly.
>
> * **Iteration 5 (Breakthrough & Correct Rationale):** The model finally assembled the correct logic.
>     1.  **Corrects the Fact:** It decisively stated that Michigan **lost** the 2016 game, correcting its initial misinterpretation.
>     2.  **Switches Evidence:** Pivoted to new documents indicating Michigan’s all-time series lead through 2018 and Ohio State’s vacated 2010 win.
>
> This showcases AlignRAG’s critique-refinement loop, where the model iteratively recognizes and corrects its faulty logic, culminating in the correct answer.
>
> **Case Study 2: Synthesizing Multiple Truths into a Coherent Whole**
>
> * **Question:** "what was the tower of london originally used for"
> * **Final Correct Answer:** "as a royal residence"
>
> **Rationale Evolution:**
>
> * **Iteration 1-4 (Over-focusing):** The model emphasized the Tower’s role as a military fortress, citing its construction post-Norman Conquest to control London. This was factually correct but incomplete.
> * **Iteration 5 (Breakthrough & Synthesis):**
>     1.  **Integrates Multiple Functions:** It acknowledged its primary defensive purpose but gave equal weight to the fact that it was also built as **"a grand palace"** and **"served as a royal residence"** from early in its history.
>     2.  **Refines the "Purpose":** It understood that for a medieval king, a fortress and a palace were not mutually exclusive; they were integrated parts of projecting power and governance. The Tower was designed from the start to be both a secure stronghold *and* a royal home.
>
> This example highlights AlignRAG’s ability to move beyond isolated facts and perform holistic synthesis—essential even in single-hop settings when multiple correct truths must be weighed.
>
> That said, `despite the simplicity of the query, the model's reasoning faltered due to contextual noise`. AlignRAG’s iterative critique acted as a debugging mechanism, realigning the rationale with both the query and the evidence.
>
> ## Quantitative Impact of Iterative Refinement
>
> To quantify the impact of these refinements, we tracked the number of **newly corrected samples at each specific iteration** across all seven benchmarks.
>
> | Dataset | @ Iter 1 | @ Iter 2 | @ Iter 3 | @ Iter 4 | @ Iter 5 |
> | :--- | :---: | :---: | :---: | :---: | :---: |
> | **PopQA** | 73 | 7 | 4 | 4 | 0 |
> | **TriviaQA** | 562 | 103 | 43 | 30 | 28 |
> | **NaturalQuestions** | 268 | 43 | 31 | 21 | 17 |
> | **2WikiMultiHopQA**| 703 | 151 | 68 | 52 | 55 |
> | **ASQA** | 67 | 8 | 4 | 2 | 2 |
> | **HotpotQA** | 1147 | 87 | 25 | 37 | 24 |
> | **SQUAD** | 1798 | 165 | 78 | 53 | 40 |
>
> In summary, `this mechanism is effective for both complex multi-hop and seemingly simple single-hop questions where retrieval quality or initial reasoning is imperfect.`

---

> ### Author Response · Authors · 2025-08-07
> **Reply 2**
>
> > *Moreover, beyond a few case studies, I believe a quantitative analysis is needed to show how and to what extent each category of RAG misalignment was improved by the proposed refinement.*
>
> We thank the reviewer for this constructive point and we fully agree on its importance.
>
> However, producing such a fine-grained breakdown would `require a significant manual annotation effort`. Each case of a corrected error would need to be carefully examined and classified into one of the three misalignment phases. This process is unfortunately not feasible within the rebuttal period, but `we acknowledge the importance of this direction for future work`.
>
> To address the spirit of the reviewer's request and `provide a more holistic quantitative measure` of our method's corrective power, we conducted a new analysis. We measured the **Correction Success Rate**: the percentage of instances where an initially *incorrect* rationale was successfully corrected to a *correct* one through refinement.
>
> We compared AlignRAG against the Self-Refine baseline using the Qwen-7B-instruct model. The results are presented below:
>
> | Correction Success Rate (%) | PopQA | TriviaQA | NQ | 2WikiMultiHopQA | ASQA | HotpotQA | SQUAD | **Average** |
> | :---------------------------- | :---: | :---: | :---: | :---: | :---: | :---: | :---: | :---: |
> | Self-Refine | 13.6 | 16.5 | 16.5 | 5.7 | 24.4 | 10.2 | 14.8 | **14.5** |
> | **AlignRAG** | **15.6** | **23.7** | **23.1** | **12.7** | **30.0** | **20.6** | **21.1** | **21.0** |
>
> As the table shows, `AlignRAG demonstrates a consistently and significantly higher success rate in correcting initially flawed reasoning across all seven benchmarks.`
>
> While this analysis does not separate the three sub-categories of misalignment, it directly quantifies our core claim: training a dedicated CLM on contrastive critiques produces superior feedback that more effectively guides the generator toward evidence-aligned and correct responses. This directly addresses the critical problem of reasoning misalignment in a measurable way.
>
> `We hope this helps further clarify the method and address your concern based on our above clarification. Thanks again for your support!! Best, authors`

---

### Official Review · Reviewer_Kguz · 2025-06-30

**Clarity:** 3
**Significance:** 3
**Originality:** 3
**Rating:** 4
**Confidence:** 4

**Summary:**

Retrieval-augmented generation (RAG) enables knowledge-grounded LLMs but frequently exhibits reasoning misalignment, where inference drifts from retrieved evidence and yields factual errors. AlignRAG reframes RAG as Retrieval-Augmented Reasoning and embeds a Critique-Driven Alignment loop in which a dedicated Critic Language Model (CLM)—trained via contrastive critique synthesis to distinguish evidence-aligned from misaligned reasoning—generates targeted critiques at inference that steer the generator back to relevant documents . Without altering existing RAG pipelines, an 8B CLM delivers substantial gains over prior refinement methods and consistently outperforms a 72B baseline, all while maintaining robustness under noisy retrieval conditions.

**Questions:**

- What are the results of runing time？
- How does AlignRAG’s performance compare with that of IRCOT [1]？

[1] Trivedi H, Balasubramanian N, Khot T, et al. Interleaving retrieval with chain-of-thought reasoning for knowledge-intensive multi-step questions[J]. arXiv preprint arXiv:2212.10509, 2022.

**Ethical Concerns:**

["NO or VERY MINOR ethics concerns only"]

**Final Justification:**

The response has adequately addressed my concerns.

**Limitations:**

If the retrieval quality is insufficient, it will negatively impact AlignRAG’s performance.

**Quality:**

2

**Strengths And Weaknesses:**

Strengths:
- Principled Framework Design: Embedding the Critic Language Model into an iterative inference loop transforms RAG into a constrained optimization process. This design elegantly marries retrieval and generation phases and provides a clear mechanism for evidence-driven correction.
- Plug-and-Play Compatibility: ALIGNRAG integrates seamlessly with existing RAG architectures without requiring modifications to the generator or retrieval components. This design choice maximizes practical applicability and lowers barriers to adoption.
- Robustness: Experimental results on out-of-domain and noisy retrieval scenarios substantiate claims of improved stability, an essential quality for real-world deployments

Weakness:
- This paper lacks of the results of runing time. The iterative critique–generation loop incurs substantial computational overhead. Multiple forward passes through both CLM and generator can increase inference time, limiting usability in latency-sensitive contexts.
- Effective deployment requires setting several hyperparameters, such as number of critique iterations, etc.—which may vary significantly across tasks, models and domains. Does this approach also require preparing a dedicated training dataset for each domain?
- Table 1 shows that AlignRAG underperforms InstructRAG, yielding even lower scores. In Table 2, combining InstructRAG with test-time  alignment yields better performance—but by how much does the cost increase?

---

> ### Author Rebuttal · Authors · 2025-07-30
>
> We sincerely thank you for your insightful suggestions. Below, we provide a point-by-point response to your concerns. **Note: The results below correspond to a single iteration of AlignRAG due to time constraints.**
>
> > *Effective deployment requires setting several hyperparameters, such as number of critique iterations, etc.—which may vary significantly across tasks, models and domains. Does this approach also require preparing a dedicated training dataset for each domain? What are the results of runing time？*
>
> We appreciate this thoughtful observation regarding the practicality and usability of the framework. Your comment directly inspired the development of a more autonomous and robust variant: `AlignRAG-auto`. This version eliminates the need for manual iteration tuning and demonstrates strong generalization without requiring domain-specific data.
>
> **Training for Dynamic Refinement:** We modified the training process for our Critic Language Model (CLM). For each initial "unexpected" response (`y_unexp`), we first determine its correctness. Incorrect responses are labeled as `[Bad]`, while correct ones are labeled as `[Good]`. The CLM is then trained to generate the appropriate `[Good]` or `[Bad]` token followed  by critique.
>
> This enables AlignRAG-auto to operate `dynamically at inference time`: it evaluates each response and terminates the refinement loop as soon as the CLM emits a `[Good]` token, saving substantial compute. This dynamic stopping mechanism removes the need for pre-defined iteration counts and makes the system more efficient and hands-free.
> `As the table below illustrates, this approach yields high accuracy while refining only a subset of samples.`
>
> **Refinement Ratio for AlignRAG-auto (% of samples requiring refinement)**:
> | Model| PopQA| TriviaQA | NQ | 2WikiMultiHopQA | ASQA | avg|
> | :--------------------|:------|:--------|:------|:----------|:------|:------|
> | AlignRAG-auto-8B | 63.0 | 50.1 | 53.3 | 56.6 | 54.8 | 55.5 |
> | AlignRAG-auto-7B | 66.3 | 49.7 | 56.9 | 55.1 | 55.7 | 56.8 |
> | AlignRAG-auto-14B | 61.0 | 47.7 | 49.5 | 55.0 | 49.9 | 52.6 |
>
> **Performance Comparison**:
> | Method| PopQA | TriviaQA | NQ| 2WikiMultiHopQA | ASQA | avg|
> |--------------------|-------|----------|-------|------------------|-----------|--------|
> | AlignRAG-8b | 66.5  | 77.0  | 65.3  | 47.0   | 47.1   | 60.4  |
> | AlignRAG-7b | 68.4  | 77.8 | 65.9  | 49.5   | 48.9   | 62.1  |
> | AlignRAG-14b | 68.4  | 79.5  | 67.7  | 49.8   | 48.6   | 62.8  |
> | AlignRAG-auto-8b | 67.6 | 77.6 | 66.8 | 47.6  | 48.8  | 61.3 |
> | AlignRAG-auto-7b | 68.1 | 78.1  | 67.3 | 49.3  | 49.6  | 62.5  |
> | AlignRAG-auto-14b | 68.3 | 79.9  | 69.0 | 50.2   | 49.8   | 63.0  |
>
> Importantly, AlignRAG-auto does not require domain-specific training. The CLM is trained on a diverse dataset spanning five benchmarks, and as shown in Figure 2 and Table 6, AlignRAG demonstrates strong generalization and robustness to out-of-distribution (OOD) settings. This supports the claim that the CLM captures general principles of evidence-based reasoning and alignment, not domain-specific patterns.
>
> `We are grateful for this question, as it pushed us to develop the AlignRAG-auto variant, which we believe makes our framework more robust, efficient, and practical. We will be adding this to the final version of our paper. `
>
> > *This paper lacks of the results of runing time. The iterative critique–generation loop incurs substantial computational overhead. Multiple forward passes through both CLM and generator can increase inference time, limiting usability in latency-sensitive contexts.*
>
> Thank you for highlighting this critical consideration. We agree that computational efficiency is vital for real-world adoption.
>
> AlignRAG indeed makes a deliberate `trade-off`, prioritizing correctness and reliability—especially important in high-stakes applications (e.g., clinical or medical RAG)—at the cost of increased compute. However, your feedback prompted us to quantify and reduce this cost.
>
> As described above, AlignRAG-auto introduces a `"refine only when necessary"` approach. The CLM bypasses refinement for responses it deems correct, reducing the average computational load significantly. Across benchmarks, `only ~55% of samples require refinement`—without sacrificing accuracy.
>
> To provide concrete results, we estimated the computational cost in terms of TFLOPs for a 7B model. The table below shows that AlignRAG-auto, especially when combined with optimizations like `Prefix-caching` [1], offers a compelling balance of performance and efficiency, bringing the cost much closer to that of non-iterative methods while delivering superior accuracy.
>
> **Estimated Computational Cost (7B Model)**:
>
> | Method | Accuracy | TFlops |
> | :---------------------------- | :------------ | :---------- |
> | Vanilla RAG | 56.9 | 13.2 |
> | Self-Refine | 58.3 | 35.6 |
> | AlignRAG | 62.1 | 78.7 |
> | AlignRAG w/ Prefix cache | 62.1 | < 46.8 |
> | AlignRAG-auto w/ Prefix cache | 62.5 | < 25 |
>
> `Future Optimizations:` The computational cost can be further mitigated by standard **engineering practices**. Techniques like **Prefix-caching** (which we've estimated in our tests), **predictive decoding**[2], and **optimized inference engines** like vLLM are complementary to our approach and can substantially reduce latency in production environments.
>
> In summary, while iterative refinement carries an inherent cost, our new **AlignRAG-auto** variant offers a highly effective solution that preserves state-of-the-art performance while significantly cutting the average computational overhead, making the framework much more practical for real-world use. We will add this analysis to the appendix.
>
> [1] ChunkAttention: Efficient Self-Attention with Prefix-Aware KV Cache and Two-Phase Partition
>
> [2] Predictive Pipelined Decoding: A Compute-Latency Trade-off for Exact LLM Decoding
>
> > *Table 1 shows that AlignRAG underperforms InstructRAG, yielding even lower scores. In Table 2, combining InstructRAG with test-time alignment yields better performance—but by how much does the cost increase?*
>
> We appreciate this careful reading. In fact, this combination is exactly where AlignRAG shines.
>
> As Table 2 shows, AlignRAG adds significant value even when applied to strong training-time baselines like InstructRAG. For example, `applying AlignRAG to the Qwen2.5-14B-Instruct model yields a 9.4% gain in OOD accuracy`—a substantial improvement. This demonstrates that training-time methods alone are not sufficient to fully resolve reasoning misalignments. AlignRAG complements such models by correcting errors at inference, providing targeted refinement based on the actual retrieved evidence.
>
> With the introduction of AlignRAG-auto, prefix cache and more optimizations, this significant performance gain is now achievable with a much more manageable computational budget, as detailed in our previous response.
>
> > *How does AlignRAG’s performance compare with that of IRCOT.*
>
> This is a very relevant baseline, and we thank the reviewer for the suggestion. IR-CoT and AlignRAG represent two different, powerful strategies for improving RAG systems.
> * `IR-CoT` focuses on `improving retrieval quality` by interleaving search and generation.
> * `AlignRAG` focuses on `improving reasoning quality` by refining the final output after retrieval is complete.
>
> To address this, we ran new experiments using Qwen2.5-7B-Instruct, limiting IR-CoT to 3 iterations and top-2 documents per round:
>
> | Method  | PopQA | TriviaQA | NQ   | 2WikiMultiHop | ASQA | Average |
> | :----------------------- | :---- | :------- | :--- | :------------ | :--- | :------ |
> | IR-CoT | 51.4  | 74.1 | 58.9 | 49.6  | 38.8 | 54.6   |
> | AlignRAG-auto | 68.1 | 78.1  | 67.3 | 49.3  | 49.6 | 62.5  |
> | IR-CoT + AlignRAG | 57.5  | 79.0   | 66.9 | 56.5  | 47.0 | 61.4   |
>
> As expected, IR-CoT performs strongly on multi-hop tasks where its iterative retrieval is most beneficial. AlignRAG demonstrates superior overall performance by focusing on the crucial final step of aligning reasoning with the provided evidence.
>
> Most importantly, the frameworks are complementary. The `IR-CoT + AlignRAG-auto` result shows that our method can take the high-quality documents found by IR-CoT and further enhance the final output by correcting reasoning misalignments. This shows that test-time alignment is a valuable and distinct contribution, even for systems with advanced retrieval strategies. We will add this comparison to the paper.
>
> > *If the retrieval quality is insufficient, it will negatively impact AlignRAG’s performance.*
>
> We agree completely. Our work aims to maximize the utility of whatever information is retrieved.
>
> Our experiments in **Figure 4 ("When RAG Retrieval Falters, AlignRAG Thrives")** were designed specifically to test our framework's robustness under imperfect retrieval conditions. The results show that AlignRAG is significantly more resilient to `noisy retrieval`—a very common real-world scenario—than existing baselines. It excels at filtering out distractor information and correctly utilizing relevant snippets when they are present.
>
> However, as noted in Appendix A.1, the framework’s effectiveness diminishes in cases of extreme retrieval failure, where no relevant information is present for the critic to reason over. We will make this limitation more explicit in the revised manuscript.
>
> **We thank the reviewer for the careful and detailed reading. Please don’t hesitate to reach out with further questions!**
>
> **Thank you again for your time!**

---

> > ### Comment · Reviewer_Kguz · 2025-08-05
> > **About your response**
> >
> > Thank you for your response. It has adequately addressed my concerns, and I have updated my score accordingly.

---

> ### Author Response · Authors · 2025-08-07
> **Reply to Reviewer**
>
> Thanks again for the thoughtful suggestion and your support!!! Best, Authors

---

### Official Review · Reviewer_UcqY · 2025-07-03

**Clarity:** 2
**Significance:** 3
**Originality:** 3
**Rating:** 4
**Confidence:** 3

**Summary:**

This paper aims to improve RAG in the generation phase. It first analyzes a critical challenge in RAG, termed Reasoning Misalignment, referring to a disconnect between the model's reasoning process and the retrieved evidence. It then tries to understand the underlying cause of the problem by reconceptualizing RAG as Retrieval-Augmented Reasoning. Further, it proposes a framework named ALIGNRAG, which employs a Critic Language Model to dynamically correct misalignments during inference using retrieval-augmented critiques. Finally, extensive experiments across seven benchmark datasets and three model families are conduct to evalute the effectiveness of ALIGNRAG.

**Questions:**

Please refer to weaknesses.

**Ethical Concerns:**

["NO or VERY MINOR ethics concerns only"]

**Final Justification:**

The authors have addressed most of my concerns.

**Limitations:**

yes

**Paper Formatting Concerns:**

1. Line 205: repeated equation
2. Line 232: missing space between contents and citations
3. Line 234: grammar mistake, "revises" -> "revise"

**Quality:**

3

**Strengths And Weaknesses:**

The main strengths are summarized as follows:
1. The studied problem is interesting.

The main weaknesses are summarized as follows:
1. The analysis of the reasoning misalignment issue of RAG in Section 3 is not clear. It would be better if an intuitive example could be provided to illustrate the two characteristics of misalignment. Besides, I doubt whether there are truly three reasoning steps involved in RAG reasoning. The connection between this analysis the proposed solution seems weak.
2. The proposed method heavily rely on the construction of training data with complicated process. As shown in Equation (1), the quality label of retrived context and the supervision signal are difficult to obtain.
3. It would be better to include a case study in the experiments. The inference process is not straightforward for me.

---

> ### Author Rebuttal · Authors · 2025-07-30
>
> We sincerely thank you for your critical feedback and insightful suggestions. We agree that the analysis in Section 3 would be significantly clearer with concrete examples and that a detailed case study is essential for understanding the inference process.
>
> > *The analysis of the reasoning misalignment issue of RAG in Section 3 is not clear. It would be better if an intuitive example could be provided to illustrate the two characteristics of misalignment. Besides, I doubt whether there are truly three reasoning steps involved in RAG reasoning. The connection between this analysis the proposed solution seems weak.*
>
> This is a very fair point, and we appreciate the opportunity to clarify our thinking.
>
> The three-phase breakdown we propose—(1) Relevance Assessment, (2) Query-Evidence Mapping, and (3) Evidence-Integrated Synthesis—is not intended as a rigid procedural pipeline, but rather as a diagnostic framework for analyzing reasoning alignment in RAG systems. While models may not explicitly follow these stages, many leading reasoning-centric RAG approaches (e.g., InstructRAG [1], Self-RAG [2]) implicitly adopt similar structures to guide retrieval-augmented generation. For instance, `InstructRAG has demonstrated that structured reasoning can effectively stimulate LLM capabilities in RAG settings`. Similarly, AlignRAG has shown that such structured processes enhance both the reliability and explainability of answers—qualities that are especially important in high-stakes domains like medical and clinical RAG systems.
>
> This framework allows us to systematically identify distinct failure modes, even when the retrieval step itself is accurate. More importantly, it serves as the foundation for our proposed solution: `our Critic Language Model (CLM) is explicitly trained to recognize and generate critiques linked to failures at each of these reasoning stages.` By diagnosing whether a response suffers from weak relevance assessment, flawed evidence mapping, or poor synthesis, the CLM can guide the generation process toward more accurate and evidence-grounded outputs.
>
> To further illustrate the utility of this framework, we will include detailed case studies in the appendix that show how real-world errors correspond to these phases of reasoning.
>
> **Case 1: Misalignment in Phase 1 — Relevance Assessment**
>
> | **Aspect** | **Details** |
> |:-----------------------|:-----------------------------------------------------------------------------|
> | **Question**           | *Who is the mother of Mary in Islam?*                                       |
> | **Golden Answer**      | Hannah |
> | **Initial Response**   | "The given documents are not relevant..."                                   |
> | **Retrieved Evidence** | Document 2 explicitly states “Hannah, the mother of Mary”                    |
> | **Critic Evaluation**  | `[Bad]` — Model failed to recognize clearly relevant evidence                |
> | **Refined Response**   | "Hannah is mentioned as the mother of Mary..."                              |
> | **Failure Mode**       | **Relevance misjudgment** led to discarded correct evidence                  |
> | **Root Cause**         | Poor signal interpretation in Phase 1                                        |
> | **Takeaway**           | Even strong retrieval fails if relevance is not properly assessed            |
>
> **Case 2: Misalignment in Phase 2 — Query-Evidence Mapping**
>
> | **Aspect**  | **Details** |
> |:-----------------------|:-----------------------------------------------------------------------------|
> | **Question**           | *What name is given to a very long forward pass in American football made in desperation?* |
> | **Golden Answer**      | Hail Mary |
> | **Initial Response**   | Cited Document 1, ignored Document 5                                         |
> | **Retrieved Evidence** | Document 5 directly defines "Hail Mary pass"                                 |
> | **Critic Evaluation**  | `[Bad]` — Model missed most direct evidence                                  |
> | **Refined Response**   | Correctly identifies "Hail Mary"                                             |
> | **Failure Mode**       | **Mapping failure** — missed clear span-to-query alignment                   |
> | **Root Cause**         | Salient match ignored during evidence mapping                                |
> | **Takeaway**           | Query-to-doc matching must prioritize direct matches over vague associations |
>
> **Case 3: Misalignment in Phase 3 — Synthesis**
>
> | **Aspect** | **Details** |
> |:-----------------------|:-----------------------------------------------------------------------------|
> | **Question**           | *Who was the director of Alexander?*                                        |
> | **Golden Answer**      | Oliver Stone  |
> | **Initial Response**   | Read correct doc but failed to state answer                                 |
> | **Retrieved Evidence** | Document 5: “It was directed by Oliver Stone…”                              |
> | **Critic Evaluation**  | `[Bad]` — Info was present but omitted in output                            |
> | **Refined Response**   | “The only director identified is Oliver Stone.”                             |
> | **Failure Mode**       | **Synthesis error** — failed to integrate known evidence                     |
> | **Root Cause**         | Incomplete summarization of relevant facts                                  |
> | **Takeaway**           | Strong evidence is useless if the synthesis step fails to surface it         |
>
> [1] InstructRAG: Instructing Retrieval-Augmented Generation via Self-Synthesized Rationales
>
> [2] Self-RAG: Learning to Retrieve, Generate, and Critique through Self-Reflection
>
> > *The proposed method heavily rely on the construction of training data with complicated process. As shown in Equation (1), the quality label of retrived context and the supervision signal are difficult to obtain.*
>
> We appreciate you raising this important concern about the complexity of our training data construction. We recognize the need to be clearer on this point. While the process is multi-faceted, we would like to emphasize that it is designed to be **`fully programmatic and self-supervised`**, avoiding any need for expensive manual annotation beyond the ground-truth answers already present in the benchmark datasets.
>
> We will clarify how both key components of our training data are generated automatically and don't not require additional human labeling:
>
> 1.  **Automated Generation of Context Quality Labels (`c_i`):** You are correct to ask how the quality labels for retrieved context (`c_i`) are obtained. This process is fully automated. Following standard practices in RAG evaluation, we programmatically check for the presence of the ground-truth answer strings within the retrieved documents to label them for "helpfulness" and "completeness." This allows us to systematically create a diverse training corpus representing various retrieval scenarios without any manual effort.
>
> 2.  **Automated Generation of Supervision Signals (`Δy_unexp,i`):** The core supervision signal—the critique (`Δy_unexp,i`) itself—is also generated automatically via our novel **Contrastive Critique Synthesis (CCS)** mechanism. By contrasting the outputs of "weak" and "strong" models for the same input, we create a high-quality and consistent supervisory signal. This self-distillation approach is a key advantage, as it `does not require human labeling` and is more `scalable`. It also constrains the learning objective, focusing the critic on identifying evidence-based misalignments rather than stylistic preferences. Furthermore, this signal is improvable; as we show in Section 5.3 ("Superior Alignment Guidance Surpassing Strong LLM Baselines"), using a stronger model to generate these critiques further boosts performance.
>
> In summary, the entire data construction pipeline is automatic, scalable, and reproducible. We will revise Section 4.2.1 to more clearly articulate these automated steps and allay any concerns about the practicality of our method.
>
> > *It would be better to include a case study in the experiments. The inference process is not straightforward for me.*
>
> This is an excellent suggestion. To make our `critique-and-refine` loop clear, we will add the following step-by-step case study to the main paper.
>
> **Query:** "what is the shape of the ohio flag called?"
>
> **Retrieved Evidence:** Multiple documents stating the flag is a "non-rectangular," "swallowtail," and "burgee."
>
> **Step 1: Initial Generation (Weak Rationale)**
> The generator model first produces a flawed rationale:
> > *"The documents are not relevant. They talk about Ohio's flag, but not its shape. "*
>
> This is a **Phase 1 (Relevance Assessment)** failure. The model has dismissed highly relevant documents.
>
> **Step 2: Critique Generation**
> AlignRAG's Critic Language Model (CLM) takes the query, evidence, and the weak rationale as input and produces a targeted critique:
> > *"The critique for the rationale is: [Bad]. The weak rationale fails to acknowledge clearly stated, relevant evidence. It dismisses all documents as irrelevant even though they explicitly mention the flag's shape, using terms like 'swallowtail' and 'burgee' ..."*
>
> **Step 3: Refined Generation**
> Guided by the critique, it produces a corrected, evidence-aligned rationale:
> > *"Documents 2–5 consistently describe the Ohio flag as a 'swallowtail' and a 'burgee,' both of which are names for a non-rectangular or V-shaped flag. Therefore, the shape of the Ohio flag is called a swallowtail."*
>
> ---
>
> > *Paper Formatting Concerns: Line 205: repeated equation. Line 232: missing space between contents and citations. Line 234: grammar mistake, "revises" -> "revise"*
>
> Thank you for spotting these. We will correct them in the next version.
>
> **Please let us know if you have any further questions—we’d be happy to address them. Thanks again for your time and thoughtful review!**

---

> > ### Comment · Reviewer_UcqY · 2025-08-04
> > **About your reponse**
> >
> > Thanks for your detailed response. I have raised my rating to 4.

---

> ### Author Response · Authors · 2025-08-07
> **Reply to Reviewer**
>
> Thank you again for your confirmation on our work and your precious time!!

---

### Note · Authors · 2025-08-12

We sincerely thank the Area Chair and all reviewers for their thoughtful and constructive feedback.

We are grateful that the reviewers `recognized the core strengths` of our work, particularly: (1) the identifying `"Reasoning Misalignment"` as a core RAG problem; (2) our innovative Contrastive Critique Synthesis (CCS) `training method` for mitigating self-bias; (3) the principled and `plug-and-play design` of AlignRAG; and (4) the empirical `rigor and robustness` demonstrated across seven benchmarks.

---

The most pointed weaknesses centered on two key areas: **1) a need for deeper qualitative insight** into our method's mechanics (UcqY, sEZn), and **2) the computational overhead** of our iterative approach (Kguz, pwHC). We have diligently addressed both through new analyses and experiments during the rebuttal period.

To address the call for clarity, we provided `detailed case studies` that illustrate not only the three phases of reasoning misalignment but also the step-by-step process of how ALIGNRAG corrects flawed rationales. This provides the tangible examples that reviewers requested.

Prompted by reviewer Kguz's feedback, we developed `AlignRAG-auto`, a more efficient variant that dynamically learns when to terminate the refinement loop. This innovation, inspired by reviewer feedback, preserves state-of-the-art accuracy while `reducing the average computational cost by nearly 50%`, making our framework far more practical for real-world use. We have added a full analysis of this, including TFLOPs comparisons.

Additionally, we expanded our empirical validation by adding new comparisons against strong baselines like `IR-CoT`, including `InstructRAG` in our analytical figures for a fairer comparison, and demonstrating `generalization to new tasks` like summarization and medical QA.

---

For the final version, we commit to making the following improvements:
* **Add extensive qualitative analysis**, including new case studies in the appendix.
* **Introduce and detail** `AlignRAG-auto`, our more efficient and practical framework variant.
* **Integrate all new experimental results**, including the `IR-CoT` comparison and generalization results on summarization and medical QA tasks.

---

Thank you again for your time and guidance. We are confident that these revisions substantially address the reviewers' concerns and that the improved manuscript represents a significant and valuable contribution to the RAG paradigm.

---

### Decision · Program_Chairs · 2025-09-17

**Decision:**

Accept (poster)

**Comment:**

The paper introduces AlignRAG that reconceptualizes RAG as Retrieval-Augmented Reasoning. The central problem identified is Reasoning Misalignment — a failure where model reasoning drifts from retrieved evidence. To address this, the authors propose a critique-driven alignment loop. The critique model generates rationales and iteratively refines them to better align with evidence. Experiments across 7 benchmarks and three LLM families show that AlignRAG improves factual accuracy, robustness to noisy retrieval, and even outperforms larger models.

All reviewers agreed that the problem is important and the solution is technically sound. Reviewers praised the conceptual novelty (pwHC, Kguz), practical design (plug-and-play, modular), and robustness results. During rebuttal, the authors provided concrete case studies (addressing UcqY, sEZn), runtime and efficiency analyses including AlignRAG-auto (addressing Kguz, pwHC), and new baseline comparisons (addressing sEZn, Kguz, pwHC).

Most concerns were addressed during rebuttal and all reviewers agree the paper has meet the acceptance bar.